# Randomized control trial of moderate dose vitamin D alters microbiota stability and metabolite networks in healthy adults

Madhur Wyatt,[1] Ankan Choudhury,[2] Gabriella Von Dohlen,[2] Jeffery L. Heileson,[1,3] Jeffrey S. Forsse,[1,4] Sumudu Rajakaruna,[5,6] Manja Zec,[5,7] Malak M. Tfaily,[5,6] Leigh Greathouse[2,4]

**ABSTRACT** Evidence indicates that both vitamin D and the gut microbiome are involved in the process of colon carcinogenesis. However, it is unclear what effects supplemental vitamin $D_3$ has on the gut microbiome and its metabolites in healthy adults. We conducted a double-blind, randomized, placebo-controlled trial to identify the acute and long-term microbiota structural and metabolite changes that occur in response to a moderate dose (4,000 IU) of vitamin $D_3$ for 12 weeks in healthy adults. Our results demonstrated a significant increase in serum 25-hydroxy-vitamin D (25(OH)D) in the treatment group compared to placebo ($P < 0.0001$). Vitamin $D_3$ significantly increased compositional similarity ($P < 0.0001$) in the treatment group, and enriched members of the Bifidobacteriaceae family. We also identified a significant inverse relationship between the percent change in serum 25(OH)D and microbial stability in the treatment group ($R = -0.52$, $P < 0.019$). Furthermore, vitamin $D_3$ supplementation resulted in notable metabolic shifts, in addition to resulting in a drastic rewiring of key gut microbial-metabolic associations. In conclusion, we show that a moderate dose of vitamin $D_3$ among healthy adults has unique acute and persistent effects on the fecal microbiota, and suggest novel mechanisms by which vitamin D may affect the host-microbiota relationship.

**IMPORTANCE** Preventative measures to reduce the rise in early-onset colorectal cancer are of critical need. Both vitamin D, dietary and serum levels, and the gut microbiome are implicated in the etiology of colorectal cancer. By understanding the intimate relationship between vitamin D, the gut microbiome, and its metabolites, we may be able to identify key mechanisms that can be targeted for intervention, including inflammation and metabolic dysfunction. Furthermore, the similarity of vitamin D to cholesterol, which is metabolized by the gut microbiome, gives precedence to its ability to produce metabolites that can be further studied and leveraged for controlling colorectal cancer incidence and mortality.

**KEYWORDS** vitamin D, colon cancer, early-onset colorectal cancer, gut microbiome, microbiome stability, inflammation

Vitamin D is involved in many extra-skeletal functions that regulate intestinal physiology and protect from disease: controlling intestinal epithelial barrier function, intestinal stem cells, inflammation, and the immune system (1–4). Large prospective epidemiological studies have reported significant associations between vitamin D deficiency and higher incidence and/or mortality of colorectal cancer (CRC) (3, 4); however, randomized control trials of vitamin D supplementation for the primary prevention of CRC have not shown a benefit. Yet early-onset colorectal cancer incidence, which is on the rise, demonstrates strong inverse associations with total vitamin D intake and serum levels of 25-hydroxy-vitamin D (25(OH)D) in large prospective cohorts,

Address correspondence to Leigh Greathouse, leigh_greathouse@baylor.edu.

We are deeply saddened to report that Dr. Forsse, a co-author of this work, passed away prior to the publication of this manuscript. We honor his invaluable contributions to this research and his lasting impact on the field.

The authors declare no conflict of interest.

See the funding table on p. 21.

suggesting a possible link with its etiology (5). The objective of this study was to investigate the microbiota as a possible factor in the relationship between vitamin D and the risk of early-onset CRC by first understanding the relationship between vitamin D intake and effects on the fecal microbiota in healthy adults.

A growing body of evidence from epidemiological and mechanistic studies indicates that the protective role of vitamin D in CRC development is exerted both through direct and indirect mechanisms (6). Specifically, vitamin D exerts direct effects on the gastrointestinal tract by modulating vitamin D receptor (VDR) expression (increased expression in early stages of CRC and decreased expression in late-stage CRC) (7) and expression of the vitamin D activation enzyme CYP27B1 (increased) during different stages of CRC progression (8, 9). These alterations in VDR and CYP27B1 can lead to downstream changes in genes and signaling pathways that can influence processes including tumor proliferation, differentiation, apoptosis, and migration (6, 10, 11). One of the indirect effects of vitamin D is hypothesized to be through modulation of the gut microbiome (12). A genome-wide association study ($N$ = 1812) demonstrated associations between *VDR* gene alterations and changes in microbiota characteristics (composition and diversity), indicating a role of vitamin D in influencing the human gut microbiota (13). In an intestinal *Vdr*-deficient murine model of azoxymethane/dextran sulfate sodium (AOM/DSS)-induced colon cancer, changes were observed in the microbiota composition that were associated with increased colon tumorigenesis (14). Conversely, human studies have demonstrated an enrichment of beneficial genera and a reduction in intestinal inflammation in response to vitamin D supplementation (15–18).

Previous human studies analyzing the effects of vitamin D on gut microbiota have observed changes in the distal gut and fecal microbiota including α-diversity, β-diversity, and taxonomic changes before and after the intervention (15, 16, 19). An observational study demonstrated that greater α-diversity (Faith's phylogenetic diversity) was associated with higher serum levels of 25(OH)D in older men [$\bar{x}$ = 84 years (SD = 4.1)] (19). However, in a 16-week randomized clinical trial of 4,000 IU vitamin D$_3$ supplementation among overweight or obese adults no significant changes in richness or α-diversity (Chao-1, Shannon index) between treatment and placebo groups were observed (15). Furthermore, previous studies have shown significant associations between community composition (β-diversity) and vitamin D, with treatment groups becoming more similar to each other suggesting an evident influence of vitamin D on gut microbiota structure (15). Several intervention studies have also identified specific taxa that are altered after vitamin D supplementation, including enrichment of *Bifidobacterium, Lachnospira*, *Akkermansia, Lactococcus,* and *Firmicutes*, and depletion of *Blautia, Lactobacillus, Ruminococcus, Bacteroides, Faecalibacterium, Proteobacteria,* and *Gammaproteobacteria* (15–17, 19–21). Again, these results were mixed, likely due to differences in dose, duration, and population.

Investigating both compositional and metabolic changes within the microbiota is crucial to gaining a more comprehensive understanding of gut microbial dynamics upon vitamin D metabolism, as metabolites represent key small molecule intermediates and end products of microbial processes. By complementing microbial composition analyses with untargeted metabolomics, additional layers of metabolic shifts and microbiota-host crosstalk influenced by vitamin D may be uncovered. However, studies that directly measure such variations in the gut metabolome after vitamin D$_3$ intervention have not been published. Hence, we conducted a randomized controlled trial of supplemental vitamin D to identify the acute and long-term microbiota structural and metabolite changes that occur in response to a moderate dose (4,000 IU) of vitamin D$_3$ for 12 weeks in healthy adults to gain insight into how vitamin D may influence the microbiome to confer benefits for protection from early-onset CRC.

## MATERIALS AND METHODS

### Study design and participant details

All study participants self-reported as healthy with no underlying health conditions. The exclusion criteria for the study included antibiotic and probiotic use in the past 2 weeks, supplemental vitamin D use in the past 2 months, self-reported or pre-existing history of inflammatory bowel disease, heart disease, or diabetes, under 18 years of age, exposure to tanning or had sun exposure of >60 min at a time in the past 4 weeks, pregnant or breastfeeding women, chronic therapy (phenobarbital, carbamazepine, spironolactone) or steroid usage within the past 2 weeks, and severe allergy to the ingredients found in vitamin D/placebo supplements. A list of drugs taken by participants is listed in Table S1. Sample size estimation was determined using an approach similar to Johnson et al. (22), except vitamin D was used as the exposure. This analysis indicated that a sample size of 20 subjects per group was sufficient to identify a change of more than 1.24 standard deviations in the top 10 microbial taxa.

After considering all inclusion and exclusion criterian listed above, we enrolled 43 participants (26 females, 16 males), aged 18–53 years into the study, vitamin D microbiota placebo-control trial (VDMT). Randomization was conducted using the participants' baseline levels of serum 25(OH)D at study day 0 after enrollment so that no significant difference in vitamin D levels confounded the group comparisons. Specifically, participants' levels of serum 25(OH)D were ordered by lowest to highest levels and then split randomly such that the matched pair had the smallest difference in baseline 25(OH)D levels. This resulted in 21 participants being randomly assigned to receive vitamin D ($n = 21$) and 22 being assigned to placebo ($n = 22$) groups. Participants were instructed to daily consume (with the largest meal) four vitamin $D_3$ gummies for 12 weeks to provide a total of 4,000 IU per day or placebo gummies. Forty-two participants completed the 12-week trial. A CONSORT (Consolidate Standards of Reporting Trials) flow diagram of VDMT is shown in Fig. 1. Due to medical reasons, one participant in the vitamin D group withdrew from the study. Separately, one participant failed a blood draw at the beginning and end of the study; thus, we used the data from 41 participants for analysis at the end of 12 weeks. The baseline demographic characteristics (weight, body mass index, lean body mass, percent body fat, and serum vitamin D levels) of the participants between placebo and treatment groups were not significantly different (Table 1).

### Supplement intervention

The vitamin D group received 4,000 IU of vitamin $D_3$ (cholecalciferol) per day for 12 weeks. This dose was set since the objective of the VDMT was not to replete clinical deficiency but to determine if the Recommended Daily Allowance upper limit of Vitamin $D_3$ (4,000 IU) could alter the microbiota in healthy individuals. The vitamin $D_3$ supplementation that was provided to the subjects was obtained from Nordic Naturals (Watsonville, CA), a third party certified for superior manufacturing procedures. Each gummy (wild berry flavor) in the container was 1,000 IU, and therefore, each participant was required to consume four gummies per day to meet the upper limit (4,000 IU) of vitamin $D_3$. The participants were instructed to consume the gummies at their largest meal to ensure maximum absorption (23). The participants who were assigned to the placebo group received vegan fruit gummies from Yum Earth Organic (Stamford, CT). This gummy fruit snack was similar in composition to the vitamin $D_3$ gummies with the exception of vitamin $D_3$, sugars (tapioca syrup and rice syrup), food additives (fumaric acid and ascorbic acid), and other ingredients (sodium citrate dihydrate, sunflower oil, and carnauba wax). A list of all ingredients for vitamin $D_3$ and placebo gummies is provided in Table S2.

# Vitamin D microbiome Trial (VDMT)

## CONSORT Flow Diagram

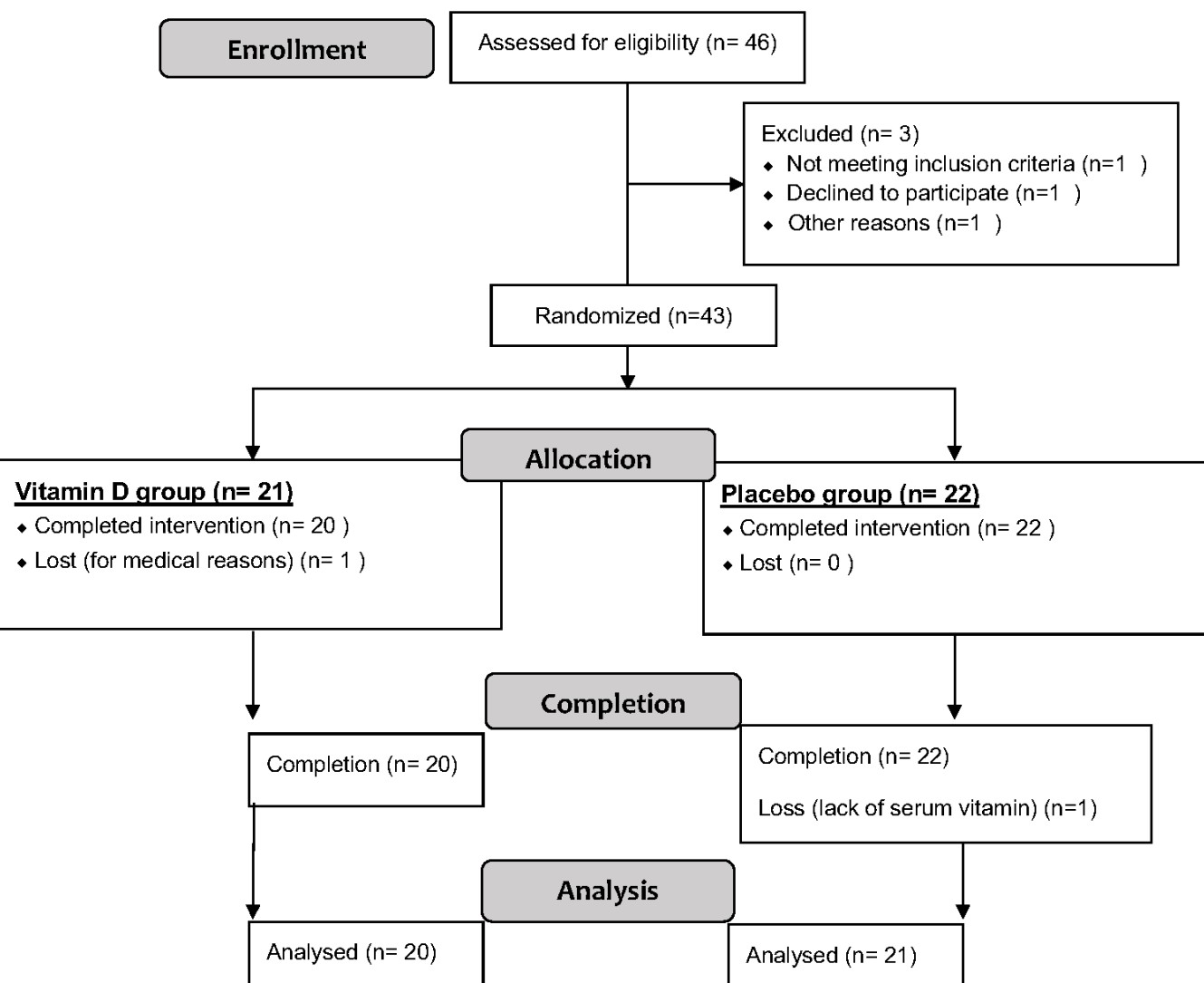

**FIG 1** CONSORT flow diagram of VDMT.

## Specimen collection and processing

### Blood sampling

Peripheral blood was collected from each participant after overnight fast on study day 1 and study day 78. Each sample was obtained by venipuncture from the most prominent vein site in the antecubital space with a 22-gauge vacutainer needle. Serum samples were collected in a red-top vacutainer tube (10 mL) and were allowed to clot at room temperature prior to centrifuging. The red-top tubes were centrifuged at 3,500 RPMs for 15 min, and serum was recovered from and allocated into 2 mL storage tubes. All serum

**TABLE 1** Participant characteristics at baseline and post-intervention[b]

| Baseline characteristic | Placebo (baseline) Mean ± SD | Treatment (baseline) Mean ± SD | P-value | Placebo (post-treatment) Mean ± SD | Treatment (post-treatment) Mean ± SD | P-value |
|---|---|---|---|---|---|---|
| Age | 24.86 ± 10.93 | 20.95 ± 3.88 | 0.130 | | | |
| Vitamin D (ng/mL) | 36.16 ± 15.19 | 39.94 ± 15.22 | 0.437 | 34.48 ± 11.95 | 72.60 ± 27.87 | <0.001 |
| Weight (lbs.) | 150.35 ± 35.92 | 157.11 ± 44.66 | 0.604 | 151.90 ± 36.28 | 157.63 ± 45.18 | 0.663 |
| BMI | 23.16 ± 3.94 | 24.56 ± 5.39 | 0.361 | 23.35 ± 3.92 | 24.67 ± 5.48 | 0.850 |
| Lean body mass | 114.85 ± 26.28 | 116.19 ± 28.99 | 0.880 | 111.62 ± 33.66 | 116.72 ± 29.42 | 0.612 |
| Percent body fat | 23.16 ± 8.80 | 24.97 ± 9.00 | 0.524 | 23.30 ± 9.06 | 24.83 ± 9.44 | 0.606 |
| HEI score[a] | 62.76 ± 12.17 | 64.38 ± 8.91 | 0.630 | | | |
| Total physical activity[a] | 0.27 ± 0.02 | 0.41 ± 0.07 | 0.083 | | | |
| Daily sedentary hours[a] | 8.73 ± 0.62 | 8.83 ± 0.68 | 0.915 | | | |
| Average daily sun exposure[a] | 1.59 ± 0.24 | 1.85 ± 0.29 | 0.500 | | | |
| Average sleep[a] | 6.98 ± 0.20 | 7.35 ± 0.18 | 0.190 | | | |
| α-Diversity | | | | | | |
| Faith | 11.11 ± 0.68 | 12.47 ± 0.59 | 0.141 | 12.70 ± 0.73 | 11.87 ± 0.54 | 0.369 |
| Richness | 202.21 ± 12.74 | 225.15 ± 13.54 | 0.224 | 229.23 ± 13.88 | 204.89 ± 10.99 | 0.177 |
| Shannon | 6.15 ± 0.11 | 5.34 ± 0.37 | 0.049 | 5.94 ± 0.22 | 5.087 ± 0.347 | 0.047 |
| Sex (number per group)[a] | | | | | | |
| Male | 8 (36.4%) | 8 (38.1%) | 1.000 | | | |
| Female | 14 (63.6%) | 13 (61.9%) | | | | |
| Race (number per group)[a] | | | | | | |
| White | 13 (59.1%) | 13 (71.4%) | 0.770 | | | |
| African American | 2 (9.1%) | 2 (9.5%) | | | | |
| Asian | 6 (27.8%) | 3 (14.3%) | | | | |
| American Indian | 0 | 0 | | | | |
| Others | 1 (4.6%) | 1 (4.8%) | | | | |
| Ethnicity (number per group)[a] | | | | | | |
| Hispanic or Latino | 4 (18.2%) | 2 (9.5%) | 0.710 | | | |
| Non-Hispanic or Latino | 17 (77.3%) | 18 (85.7%) | | | | |
| Others | 1 (4.6%) | 1 (4.8%) | | | | |
| Baseline dietary pattern[a] | | | | | | |
| Western | 14 (63.6%) | 12 (57.1%) | 0.902 | | | |
| Prudent | 8 (36.4%) | 9 (42.9%) | | | | |

[a]These variables were taken once at the baseline.
[b]Data are presented as mean ± standard deviation. 25(OH)D levels among the treatment group after 12-week intervention are statistically significant ($P < 0.05$).[a] Vitamin D serum (25(OH)D) levels. BMI, body mass index; HEI score, Healthy Eating Index score.

samples were stored at −80°C until final biochemical analysis was completed. All blood drawings were performed by trained research personnel.

### Stool collection and processing

Using the identical protocol from the laboratory of Rob Knight ("Detailed Instructions for Sampling"), each participant provided daily stool samples for a maximum of 15 time points (day 1 to 14 and day 78) (Fig. S1). A 6-inch (15 cm) PurFlock Ultra Sterile Flocked Collection Swab (Ref- 25-3317-U BT, Puritan Medical Products Company, ME) was provided to each participant for daily (14 consecutive days) stool sample collection. To avoid contamination, participants were instructed (detailed verbal and visual paper instructions provided) to collect fecal material on the stool swabs without touching the toilet paper. Upon collection, the participants were instructed to place the swab back into the vial, seal it into a three-ply biohazard specimen bag, and freeze it until it was ready to drop off at the lab. Two stool drop-off locations were provided to participants that were temperature-controlled. Since stool collection was self-administered, a training session was provided to every participant for stool collection, storage, and drop-off. We

anticipated collecting one stool sample daily. However, if the participant did not pass a stool each day, they were required to drop off the empty stool kit. Upon receipt, the stool swabs were stored at −80°C, within 2 hours, for future 16S rRNA gene sequencing using Illumina MiSeq instrument or metabolomics.

### Body composition

Body composition using InBody 570 body composition analyzer was performed to measure weight, lean body mass, percent body fat, and body mass index for each participant. This information was obtained on study days 1 and 78 of the intervention. Height was self-reported.

### Dietary data

Dietary records were collected using Diet History Questionnaire III (DHQ III), and Automated Self-Administered 24-hour Dietary Assessment Tool (ASA24, 2020). Each participant was instructed to take the DHQ III only once at any time during the first week of the trial. Healthy Eating Index (HEI) scores were computed using the DHQ III database-provided algorithm. ASA24 dietary records were collected daily starting from day 1 to day 14 (to coincide with stool samples), and once at the end of the study at day 78. Participants were encouraged to log into the ASA24 website and enter everything they had to eat and drink (including water) for each day of ASA24 data collection. Multiple logins into ASA24 website were allowed throughout the day, making it easier to recall everything the participants consumed.

### Physical activity data collection

Participant exercise levels were assessed using Global Physical Activity Questionnaire, a validated tool to measure physical activity, even in different cultural practices (24). Not only did the questionnaire include questions that provided the amount of time spent on activity, but it also included questions on the amount of time the participant was sedentary. Both exercise and sedentary conditions have been associated with microbiome modulation and CRC prevention/initiation and progression (24–26).

### Sun exposure data collection

A UVSkinRisk Survey (adapted from 27) was used to assess the total sun exposure from different activities that the participants engaged in (daily outdoor activities: sunbathing, walking, gardening, mowing, eating, drinking, and attending sporting events) (28).

## Biochemical measures

### DNA extraction from stool samples

The collected stool samples were processed using Quick-DNA Fecal/Soil microbiome 96 Kit (Zymo Research) according to the manufacturer's protocols. The brief procedure is as follows: the sterile flocked collection swab was detached from the stalk and added to the bead-bashing tubes supplied, followed by physical disruption by bead beating (TissueLyzer II) at 30 Hz for 30 min. Then the supernatant was separated by centrifugation at 3,000 $g$ for 5 min, lysed with proprietary genomic lysis buffer, run through a silicon-based DNA filter, and purified by multiple washings. The resultant ultra-pure DNA was used for quality and quantity analysis. The DNA concentration and purity were evaluated using Quant-iT dsDNA Assay Kit (Thermo Scientific). The extracted DNA samples were stored at −20°C until library preparation. Aseptic techniques were employed throughout the procedure to reduce spurious contamination. A blank sterile water control was also used in the extraction process to further control contamination. A positive control of known bacteria and concentrations was used during DNA sequencing preparation, ZymoBIOMICS Microbial Community DNA Standard (Zymo Research).

## Serum vitamin D analysis

Frozen serum samples (before and after the intervention) were removed from −80℃ freezer and allowed to thaw and come to room temperature for analysis of vitamin D. Vitamin D (25(OH)D) levels (ng/mL) were measured using the Crystal Chem-Total 25-OH vitamin D ELISA Kit, catalog # 80987 (Crystal Chem, IL). Coefficient of variation of measured serum vitamin D levels (performed in duplicates) for all participants was below 10%, the usual industry standards.

## DNA sequencing and microbiota profiling

The 16S rRNA variable region V4 was amplified with polymerase chain reaction (PCR), using the following primers: 16S forward primer (515 F) with adapters TCG TCG GCA GCG TCA GAT GTG TAT AAG AGA CAG GTGYCAGCMGCCGCGGTAA 3´, and 16S reverse primer (926 R) with adapters 5´ GTC TCG TGG GCT CGG AGA TGT GTA TAA GAG ACAG CCGYCAATTYMTTTRAGTTT 3´. The PCR mixture comprised 1 µL of each forward and reverse primer (12.5 µM), 5 µL of extracted DNA of approximately equal concentration from each sample, 12.5 µL of 2 × Platinum Hot Smart Master Mix (Thermo Fisher Scientific), and water to make a final volume of 25 µL. The amplifications were performed under the following conditions: initial denaturation at 94℃ for 5 min, followed by 35 cycles of denaturation at 94℃ for 45 s, primer annealing at 50℃ for 1 min, and extension at 72℃ for 1 min and 30 s, with a final elongation at 72℃ for 5 min. The presence of PCR products was visualized by electrophoresis using a 1.5% agarose gel. The products were cleaned using Sera-Mag select PCR clean up kit (Cytiva). A second PCR was conducted to attach the Illumina adapters and barcode index primers I5 (forward) and I7 (reverse). After adapter and index attachment, the amplicons were normalized and pooled together in a DNA library at a concentration of 4 nM, measured by Quant-iT dsDNA Assay Kit (Thermo Scientific). The pooled DNA library was paired-end sequenced at 2 × 300 bp using a MiSeq Reagent Kit v.3 on Illumina MiSeq platform (Illumina, SD, USA), at Baylor University.

## 16S rRNA sequence data processing and statistical analysis

The sequences were evaluated, demultiplexed, and filtered using QIIME2 (version 2022.8). Both paired reads were trimmed from the forward end and read length of at least 200 bp for further processing to generate the amplicon sequence variants (ASVs). The denoising and filtering of chimeric sequences was done using the DADA2 plugin of QIIME2. The samples were rarefed at a depth of >25,000 which retained all the samples and 45% of features. Taxonomic classification was performed utilizing the QIIME2-compatible pre-trained feature classifier based on 16S rRNA gene database from Silva 16S rRNA 138 database. The resultant α-diversity, β-diversity, and ASV tables were downloaded and further analyzed in R 4.2.1 and RStudio platform 2022.07.1. Relative abundance analysis of taxa was performed using DESeq2 and ALDEx2.

## Microbiota functional profiling

Functional analysis of the microbial taxa data was performed using PICRUSt2 (version 2.4.1) (29). The input files were a FASTA file of representative sequences and a BIOM table of the abundance of each ASV across each sample from the vitamin D study. The steps of the pipeline used were (i) sequence placement, (ii) hidden-state prediction of genomes, (iii) metagenome prediction, and (iv) pathway-level predictions. The following pipeline was followed to perform this analysis: https://github.com/picrust/picrust2/wiki/Full-pipeline-script. Output predictions were based on Kyoto Encyclopedia of Genes and Genomes (KEGG) orthologs (KOs) and Enzyme Commission numbers (EC numbers).

## Liquid chromatography tandem mass spectroscopy (LC-MS/MS)

Fecal material was initially separated from each swab using sterile tweezers. Each swab tip was then proportionately cut and measured together with the separated fecal

material, which was then adjusted using a reference swab sectional to obtain the dry fecal weight of each sample. Each sample was then added into individual sterile glass vials with 1.5 mL of water, incubated for 2 min at room temperature, and sonicated for 10–15 min to dislodge all fecal material from the swab. Swab tips were removed from each sample using sterilized spatula to obtain the fecal slurry of each sample and used for downstream Folch extraction. For each sample, Folch extraction was carried out using chloroform, methanol, and dd water ratio of 8:4:3 as done before (30, 31). The methanol volume of each sample was based on the calculated fecal weight, where 6:1 methanol:fecal sample ratio was employed across all samples. This mixture was briefly vortexed and incubated at −20°C overnight to facilitate protein precipitation. Lipid containing methanol layer and the protein layer were carefully removed from each sample to obtain the layer of metabolite extracts. This metabolite layer was transferred to two 2 mL glass autosampler vials (400 µL each), dried in a vacuum centrifuge (Eppendorf Vacufuge Plus), and stored at −80°C. Samples were then reconstituted in 80:20 water:methanol (for reverse-phase C18 liquid chromatography tandem mass spectrometry, RP), and 50:50 water:acetonitrile (for hydrophilic interaction liquid chromatography, HILIC), prior to injection into the instrument. Pooled quality control (QC) sample was created by combining aliquots from the individual biological samples under investigation. Pooled QC-based area corrections were then employed to standardize ionization levels across the entire experiment and aid in the correction of various systematic biases inherent in large-scale metabolomics studies. Metabolite extracts were analyzed using liquid chromatography electrospray ionization tandem mass spectrometry (LC-ESI-MS/MS) using a Thermo Vanquish UHPLC system interfaced with a Thermo Exploris 480 Orbitrap mass spectrometer, using two different phases: RP and HILIC. Use of two different phases enabled better recovery of overall metabolites, where the former (RP) allowed a better recovery of non-polar metabolites (such as lipids or cell membrane components), while the latter (HILIC) allowed a better recovery of polar metabolites (such as sugars or amino acids). Samples were injected in a 1 µL volume column and eluted as follows: for RP, a gradient from 99% mobile phase A (0.1% formic acid in $H_2O$) to 95% mobile phase B (0.1% formic acid in methanol) over 16 min was used; for HILIC, a gradient from 99% mobile phase A (0.1% formic acid, 10 mM ammonium acetate, 90% acetonitrile, 10% $H_2O$) to 95% mobile phase B (0.1% formic acid, 10 mM ammonium acetate, 50% acetonitrile, 50% $H_2O$) was used. Both columns were run at 45°C with a flow rate of 300 µL/min. RP LC used a Waters ACQUITY Premier HSS T3 column (1.8 µm, 2.1 mm × 150 mm) with a column temperature of 45°C and a flow rate of 300 µL/min. HILIC LC used an ACQUITY UPLC BEH HILIC amide column (1.7 µm, 2.1 mm × 150 mm) with a column temperature of 45°C and a flow rate of 300 µL/min. The gradient was identical for HILIC and RP chromatography.

### Metabolite identification, quantification, and downstream analysis

Full MS scan data were acquired at a resolving power of 120,000 full width at half maximum (FWHM) at m/z 200, with a scanning range from m/z 65 to 975. The automatic gain control (AGC) target was set at 50%, with a maximum injection time of 100 ms. Data-dependent acquisition (DDA, dd-MS2) parameters used to obtain product ion spectra included a resolving power of 30,000 FWHM at m/z 200, AGC target of 50% ions with the maximum injection time set to auto, an isolation width of 1.2 m/z, and higher-energy collisional dissociation (HCD) collision energies of 20%, 40%, and 80%. The top 10 intense features were selected per MS1 for the DDA experiment. Samples were analyzed in both positive and negative ionization modes and confident metabolite identifications were made through Thermo Compound Discoverer 3.3 using previously described parameters (32). Only level 1 identified compounds (matching MS1, MS2, and retention time, all) and level 2 annotations (matching MS1 and retention time only) were used for subsequent interpretations, while all remaining features (level 4) were treated as unannotated features for all downstream metabolomic analyses (33). All detected features across RP and HILIC modes are provided elsewhere, including their abundances

(Supplementary data set, Workbook 1). Feature abundances were first standardized using the individual fecal weight of the sample, and this feature abundance per gram fecal weight was normalized using log transformation followed by mean centering as done before. Principal response curves analysis (PRC) was utilized to evaluate the specific contributions of individual metabolites to the overall temporal change in the vitamin D metabolome (34). Temporal shifts in the metabolome due to vitamin D were assessed by comparing standardized PRC scores across different groups. The line of equality served as the reference point, indicating the minimal change between the groups. Using histograms, the degree of variation in standardized PRCs scores for annotated metabolites between groups was categorized into bins, highlighting the top differentially changing metabolites—RP ($n = 72$) and HILIC ($n = 20$). Metabolite overrepresentation analysis was conducted using MetaboAnalyst 6.0 (35). Microbe-metabolite correlations were generated using debiased sparse partial correlation algorithm where only the correlations above |1.00| - |0.75| were used for subsequent co-occurrence networks (36). Co-occurrence networks were built using relative abundance of corresponding samples using two time points (day 1 and day 7) and metabolomes (day 2 and day 8). Individual co-occurrence networks were cross compared using DyNet tool on Cytoscape to infer network rewiring (37, 38).

## Statistical analysis

Baseline participant characteristics (Table 1) are expressed as means and standard deviations. Differences in participant characteristics between the groups were measured using paired $t$-test for significance ($P < 0.05$) for continuous variables or chi-squared test for categorical variables. α-Diversity (the microbiota richness and evenness of distribution in the microbial community) was measured using three methods: Shannon's diversity index, Faith's phylogenetic diversity (PD), and richness (ASV) in QIIME (Quantitative Insights into Microbial Ecology) (2022.8). Repeated measures analysis of variance (ANOVA) was used to determine significant differences ($P < 0.05$) in α-diversity between days. To better understand which distance matrix was the best fit for our data for β-diversity analysis, we performed a comparison of the most commonly used distance matrices in comparison with number of reads between sample pairs. From this analysis, it was observed that Aitchison's distance matrix was robust to the difference in number of reads per sample (Fig. S7). Thus, β-diversity was measured using Aitchison's distance, and significance between days and groups was determined using the permutational multivariate analysis of variance (PERMANOVA) test. Inverse Aitchison's distance was performed to calculate microbial stability. To compute dispersion of sample distance from the centroid, PermDisp analysis was compared between two groups by PERMANOVA using vegan's betadisper() function using samples from days 7, 14, and 78. A univariable regression analysis was performed to determine the effect of change in serum 25(OH)D levels (independent variable) on microbiota stability (dependent variable). Pairwise ANOVA, DESeq2, and ALDEx2 were performed to identify taxonomic changes (enrichment/depletion) over time. For the analysis of inferred metabolic function (PICRUSt), we modeled the difference between placebo and treatment groups using a generalized linear mixed model, using either EC, KO, or pathway % abundance modeled by placebo or treatment status (fixed effect) and participant ID (random effect). Principal response curve analysis was employed to evaluate the specific impact of individual microbes on the overall temporal shift in the microbiota of the vitamin D group (34).

## RESULTS

### Participants characteristics and effects of supplemental vitamin D₃ on serum vitamin D (25(OH)D) levels

Initially, we compared the characteristics of the intervention and placebo groups to identify any confounding factors. First, we examined our main outcome, serum 25(OH)D

change from baseline. At baseline, 40 participants had 25(OH)D levels above 20 ng/mL and two participants with 25(OH)D levels below 20 ng/mL (deficient −15.926 ng/mL and 13.711 ng/mL) (Table S3). Serum 25(OH)D levels for both groups at baseline were not significantly different (mean ± SD: 36.16 ± 15.20 ng/mL vs 39.94 ± 15.22, Table 1). Additionally, all other demographic factors tested did not show significant differences between the groups (Table 1). After 12 weeks of vitamin $D_3$ supplementation, (controlling for age, sex, race, body mass index, HEI, and dietary vitamin D intake; multivariate analysis of variance (MANOVA), $P > 0.2$), we found that serum 25(OH)D levels of the treatment group were significantly higher when compared to the placebo group (mean ± SD: 72.60 ± 27.87 vs 34.48 ± 11.95 ng/mL, $P < 0.0001$ (Fig. 2). These data demonstrate that 12 weeks of daily vitamin $D_3$ supplementation at 4,000 IU per day significantly increased serum 25(OH)D levels in the intervention group.

## Vitamin $D_3$ supplementation impacts acute and long-term fecal microbiota composition

To examine how vitamin $D_3$ intervention affects the gut microbiota composition, we conducted 16S rRNA sequencing on 164 samples at four time points; study days 1, 7, 14, and 78. Using three measures of α-diversity (Shannon, Faith's PD, richness), we examined changes between time points within the placebo and treatment group. While the treatment group did not demonstrate significant changes in any of the α-diversity measures, we observed significant differences between time points in the placebo group for measures of Shannon (days 1–7, days 1–14), Faith's PD (days 1–7, days 1–14), and richness (days 1–7) (Wilcoxon test, $P < 0.05$) (Fig. S5A through F). We also performed linear regression and Spearman correlation analysis between our participant demographic variables (Table 1) and serum 25(OH)D, change in 25(OH)D and microbiota stability, stratified by males and females, and by treatment group. Results from this analysis (Fig. S11), though underpowered, did not show strong trends with the exception of the positive relationships between baseline and post-intervention serum 25(OH)D, sitting duration and microbiota stability, and an inverse relationship between average weeknight sleep (hours) and vitamin D post-intervention.

The lack of significant changes in α-diversity measures among the treatment group as compared to the placebo group prompted us to investigate other microbiota measures further.

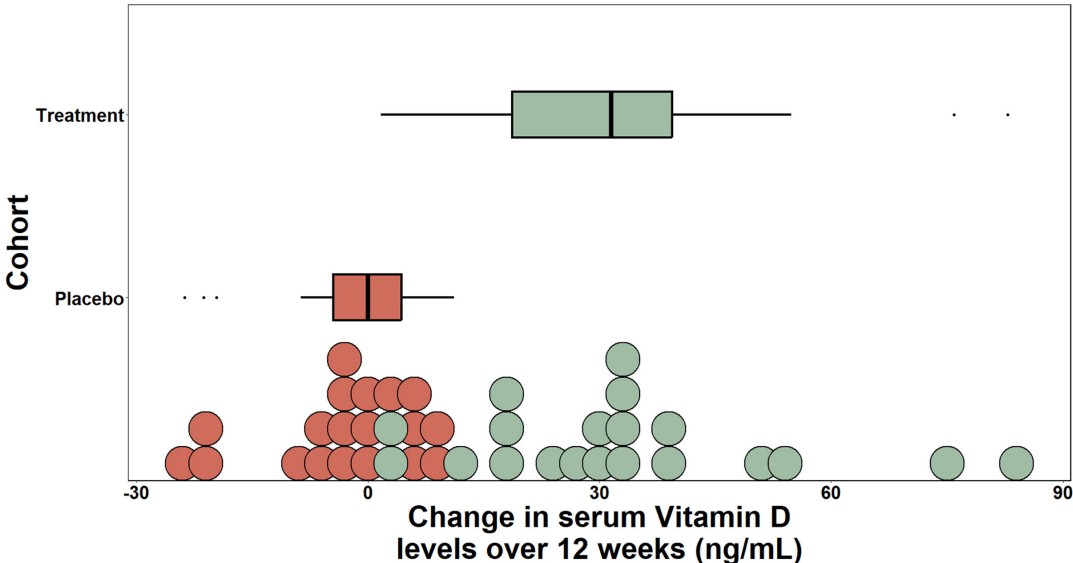

**FIG 2** Vitamin $D_3$ intervention increased serum 25(OH)D over 12 weeks. Changes in serum 25(OH)D levels over 12 weeks between placebo and treatment group measured in ng/mL. 25(OH)D levels among the treatment group after 12-week intervention are statistically significant ($P < 0.0001$).

Based on these findings, we next asked whether the microbiota stability (inverse Aitchison's distance) was maintained across time by group and whether serum 25(OH)D changes impacted this stability. Two analyses were performed: first, we calculated β-diversity (Aitchison's distance) at each time point by group (Fig. 3A and B). This analysis identified significant differences in community composition between the placebo group and the treatment group ($P < 0.0001$; paired Wilcoxon test). Within the treatment group, we observed that the amount of similarity between individuals increased over time ($P < 0.0001$, paired Wilcoxon test), which was not reflected in the placebo group. To determine how similar the microbiota composition was between groups over the course of the study, we also measured dispersion (spread or scatter of β-diversity values within a group) based on the centroid of Aitchison's distance between days 7, 14, and 78 of each participant in both placebo and treatment group. These results demonstrated that the groups were significantly different in terms of spread indicating increasing homogeneity in community composition within the treatment group after vitamin D₃ supplementation compared to placebo (false discovery rate [FDR] $P = 0.006$, PERMANOVA) (Fig. 3C). Next, we asked whether the amount of change in serum 25(OH)D levels corresponded to microbiota stability. To measure stability across changes in serum 25(OH)D, we computed the inverse of Aitchison's distance and regressed it across percent changes in serum 25(OH)D levels (22). While the participants in the placebo group did not show a correlation with stability and percent change in serum 25(OH)D, those in the treatment group demonstrated a significant inverse association between stability and increased serum 25(OH)D levels ($R = -0.052$, $R^2 = -0.27$, $P < 0.019$) (Fig. 3D). These results indicate that moderate doses of supplemental vitamin D₃ alter microbial stability among healthy individuals.

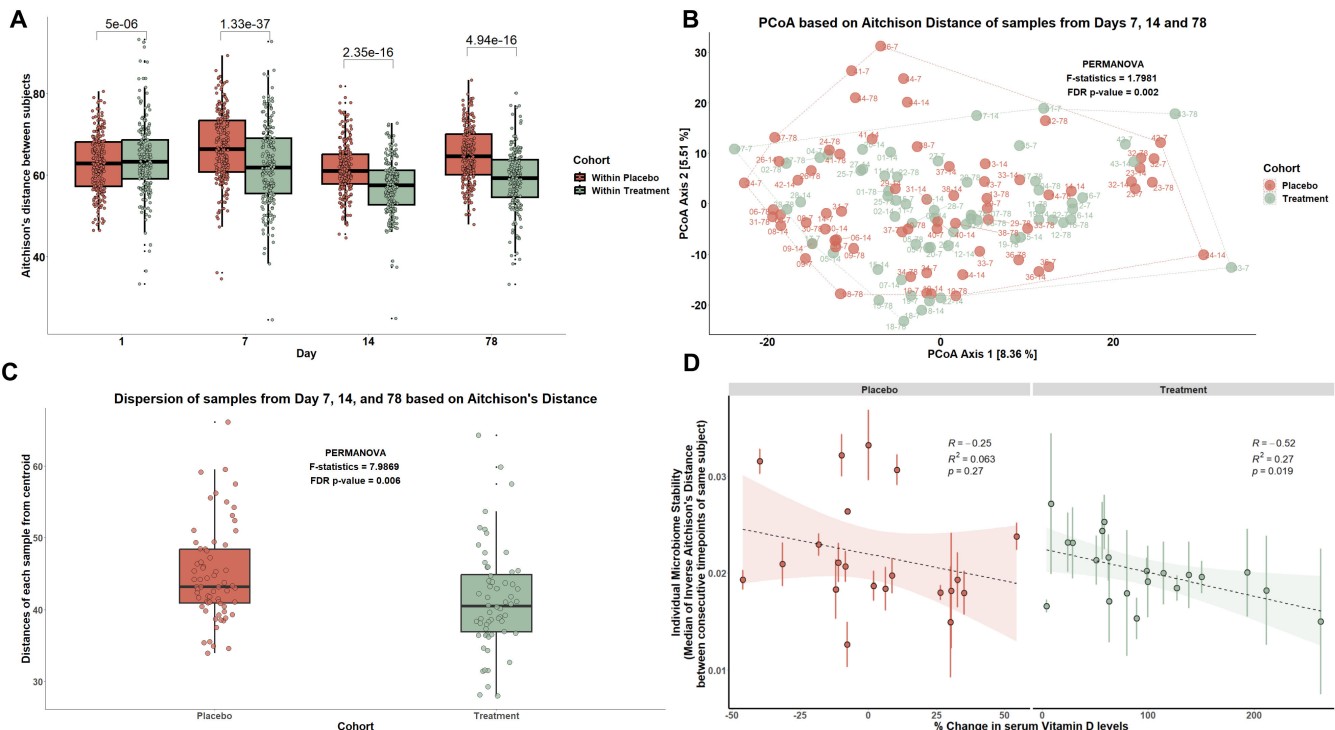

**FIG 3** Vitamin D₃ intervention alters microbiome β-diversity and homogenizes microbial composition. (A) The comparison of β-diversity using Aitchison's distance between the subjects within placebo group and within treatment group at four time points: days 1, 7, 14, and 78 (Wilcoxon's paired test). (B) Principal coordinate analysis (PCoA) comparison of community composition between groups using PERMANOVA (Adonis2, $n = 1,000$) between all samples from days 7–14 and 14–78 of each participant. (C) Dispersion analysis based on the Aitchison's distance between days 7, 14, and 78 of each participant in both placebo and treatment group (PERMANOVA, $n = 1,000$). (D). Microbiome stability (inverse Aitchison's distance) between consecutive time points for each participant in the placebo and treatment group with percent changes in serum 25(OH)D levels (generalized linear regression). Each vertical line of dots represents one participant and each dot within this line is the mean of the Aitchison's distance between days 1–7, 7–14, and 14–78.

## Supplemental vitamin D$_3$ alters specific microbial taxa in the fecal microbiota

To further our understanding of the acute and persistent effects of supplemental vitamin D$_3$ on fecal microbiota composition, we next examined microbial taxonomic abundance at the ASV level by time points and treatment groups (Fig. 4A and B). Pairwise ANOVA, DESeq2, and ALDEx2 were performed to identify taxonomic changes (enrichment/depletion) over time. The focus of this analysis was on microbial consortia or taxa associated with inflammation and CRC risk and/or prevention. Our approach began by first identifying significant changes in microbial ASVs between day 0 and days 7, 14, and 78 after vitamin D$_3$ intervention (pairwise ANOVA, DESeq2, and ALDEx2). Second, we compared these ASVs with significant changes after vitamin D$_3$ supplementation at $P <$ 0.05, and $P <$ 0.01 to the taxa that are known to either be enriched or diminished in CRC (CRC-associated microbial species) using previously published studies (Table S4). Overall, we observed 33 taxa that were significantly differentially abundant in the treatment group (Fig. 4B; Fig. S3). Among these, 17 were associated with CRC risk or prevention according to previous studies (Table S4). To better understand whether vitamin D$_3$ intervention induced compositional changes in the gut microbiota that were reflective of disease prevention or promotion, we searched the literature to explore evidence on the 17 microbes that were identified to be significantly different in the treatment group and were associated with CRC progression/prevention (Table S4). We found 12 taxa that were enriched, including *Anaerostipes, Bifidobacterium*, and *Ruminococcaceae UCG-004*, and 15 that were diminished, including *Lactococcus, Phascolarctobacterium, Ruminococcus NK4A214, Bilophila,* and *Eubacterium eligens* in the treatment group. The enrichment or depletion of these taxa was significant (FDR corrected $P <$ 0.05) for at least one time point during the study or multiple time points. To determine if these taxa also contributed to the change in β-diversity in the treatment group, as measured by Aitchison's distance from day 1 in the treatment group to subsequent days, we calculated the taxonomic abundance change that significantly correlated (Pearson's, FDR adjusted <0.05) with Aitchison's distance (Fig. S4). *Bilophila, Parabacteroidetes, Ruminococcus*, and *Lactobacillus* abundance negatively correlated (Pearson's correlation, FDR adjusted $P$-value <0.05) with the β-diversity (i.e., in subjects that had a greater change in the β-diversity compared to day 1, the abundances of these bacteria generally decreased). Whereas *Akkermansia, Coprococcus, Prevotella, Eubacterium*, and *Methanobrevibacter* abundance increased significantly in subjects whose β-diversity distance from day 1 increased in subsequent days. Next, we used a complementary method that examined the treatment group, to identify specific acute (meaning the trend was not durable over the entire study period) vs persistent (meaning the trend was durable throughout the study) taxonomic variations over time to find taxa highly responsive to vitamin D$_3$ supplementation. In addition, we assessed the persistence of these changes in the absence of vitamin D$_3$ over an extended period. To assess the global temporal shifts in the microbiota of participants in the treatment group, we used PRC analysis as previously described (34). Using the microbiota at day 1 as the baseline, we assessed the global change of gut microbiota from days 7 to 14, using Chord-transformed data. The top 4 genera that increased and decreased over time during the study in response to vitamin D$_3$ supplementation were identified through PRC taxa scores and their relative abundances were plotted along with their calculated relative abundances for day 78 (Fig. 4C). Specifically, *Bifidobacterium* and *Firmicutes* increased acutely while *Bacteroides* and *Faecalibacterium* decreased acutely. The bacteria that persisted in their change included increases in *Anaerostipes* and *Erysipelotrichaceae UCG-003,* and decreases in *Prevotella* and *Eubacterium coprostanoligenes.* Together, these data indicate that vitamin D$_3$ supplementation has both acute and persistent effects on specific taxa, including those associated with CRC.

Next, to identify taxa that are correlated with changes in serum 25(OH)D, and thus possibly involved in vitamin D metabolism, we calculated Spearman correlation coefficients using the change in relative abundance (centered log-ratio) from day 1 to days 7, 14, and 78 in the treatment group. This analysis showed a moderate-strength significant inverse relationships between the percent increase in serum 25(OH)D and

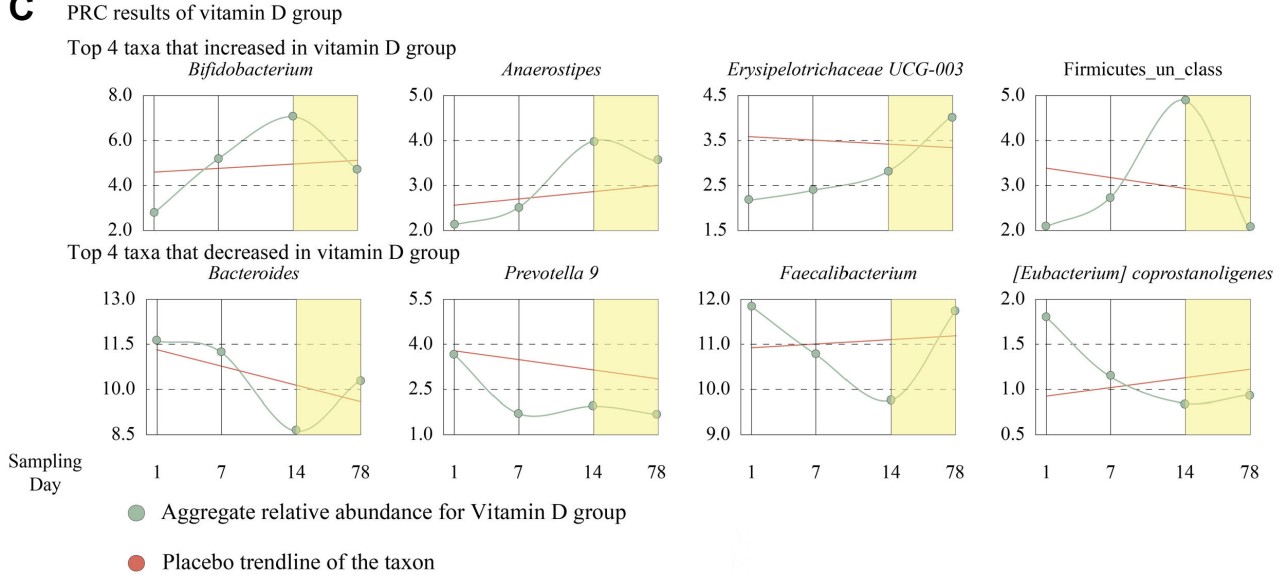

**FIG 4** Vitamin D3 induces changes in specific taxa that are both acute and persistent. Heatmaps showing changes in microbial taxonomic abundance at the genus level across the days for the placebo (A) or treatment (B) group (aggregate of features or ASVs from pairwise ANOVA, DESeq2, and ALDEx2). The three rows represent taxonomic changes from days 1–7, 7–14, and 14–78 in order. Red represents a decrease in microbial population while green represents an increase in

Fig 4 (Continued)

microbial population (*, $P < 0.05$; **, $P < 0.01$). (C) The changes in relative abundances of the top temporally changing microbes of the vitamin D group. PRCs were used to determine the contribution of each microbe toward the collective microbiome change over time in vitamin D microbiomes, where day 1 was used as the PRC baseline to compare the collective microbiota changes in days 7 and 14. PRC species scores were used to determine the top positively and negatively influenced temporal microbial changes in the vitamin D group. Y-axis represents the mean relative abundance calculated for each organism, while the x-axis depicts sampling days. In addition, the relative abundance of day 78 is added separately into each plot to showcase the persistence of these temporal changes beyond the original experimental timeframe. Placebo trendline shows the general trend of each plotted organism for the placebo group across the days 1 to 78.

*Eggerthella, Ruminococcaceae UCG-004,* and *Erysipelatoclostridium* that persisted to day 78 (Fig. 5). In contrast, we observed a positive significant relationship between serum 25(OH)D and *Gemella, Fournierella,* and *Ruminococcaceae UCG-010,* which also persisted to day 78. As a final step in our analysis, we utilized the algorithm PICRUSt2 to predict metagenomic functions in our microbiota samples to potentially identify any specific pathways being affected by microbiota exposure to vitamin D (29). Using general linearized mixed models, we compared the difference in KO, EC, or MetaCyc pathways between placebo and treatment groups at study end. After adjusting for multiple comparisons, we did not find any significantly different pathways that distinguished placebo and treatment groups (Fig. S8 to S10). Overall, these data show significant taxa-specific correlations with increases in serum 25(OH)D.

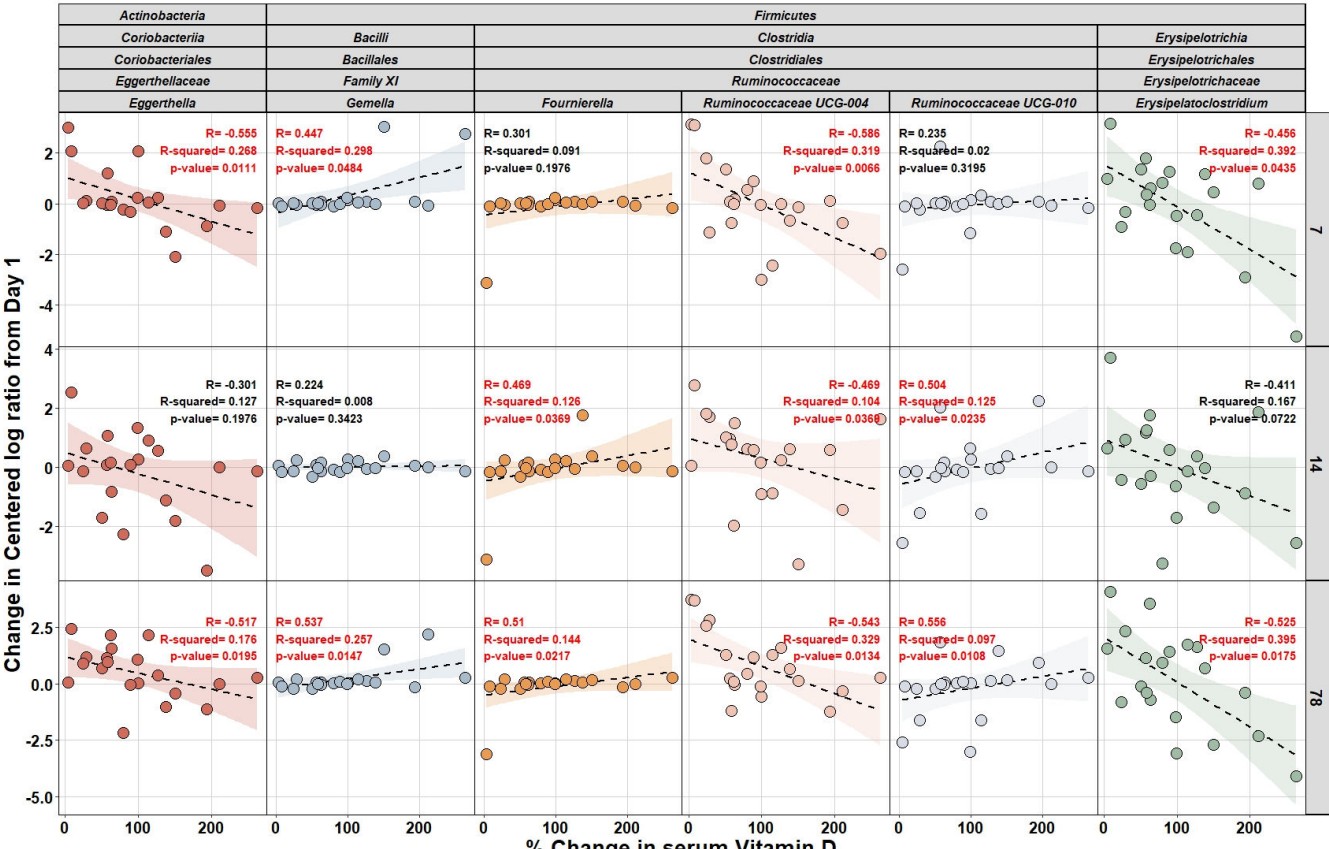

**FIG 5** Vitamin D$_3$ levels are correlated with change in the abundances of specific taxa. The graph shows the taxa with a significant correlation (Pearson's, $P < 0.05$) between the change of their abundance (centered log-ratio) for days 7, 14, and 78 from day 1 and the change in serum vitamin D level in the subjects of the treatment group.

## Supplemental vitamin D₃ modulates gut metabolome and microbe-metabolite associations

Lastly, we asked whether we could identify any key metabolite changes in the fecal microbiota that could provide clues to vitamin D metabolism by the microbiota. To decipher parallel shifts in the fecal metabolome, we utilized PRC to assess temporal variation of each metabolite with vitamin D treatment versus placebo. Species scores were standardized and compared between groups (Fig. 6A). Plot quadrants were used to easily visualize relative trends—perpendicular distance from the equality line shows the magnitude of change [the complete set of all metabolites and their relative contributions across each data set can be found in supplementary data [Workbook 1]]. Among RP metabolites, methionine, acetylcholine, tryptophan, and glutamine displayed consistent temporal increases with vitamin D₃ supplementation, contrasting minimal or opposing trends observed with placebo (Q1). Among these metabolites, the top differential contributors RP ($n = 72$) and HILIC ($n = 20$) were selected for pathway overrepresentation analysis. These 92 metabolites exhibited enrichment in pathways such as D-amino acid metabolism, arginine and proline metabolism, as well as phenylalanine tyrosine and tryptophan biosynthesis (Supplementary data, Workbook 1). In addition, these metabolites also showed significant correlations with key taxa within the treatment group (Supplementary data, Workbook 1).

Having found significant changes in both the overall gut metabolome and microbiota due to vitamin D supplementation, we further investigated how these interact (Fig. 6B). We constructed co-occurrence networks using highly correlated microbe-metabolite pairs (|0.75| and higher) separately for the placebo and treatment groups. In the placebo group, the network contained 1,931 nodes and 3,490 edges (Fig. 6B_C, 0.27% of total correlations), while the treatment group network comprised 1,879 nodes and 1,704 connections (Fig. 6B_D and D, 0.14% of total correlations). DyNet analysis revealed that, although most top correlated nodes were common between the two groups (86%, Fig. 6B_A), the interactions differed significantly (99.8%, Fig. 6B_B). These findings strongly support the importance of understanding metabolic changes beyond the changes in taxonomic abundance, underscoring the impact of microbial metabolism shifts following vitamin D supplementation.

## DISCUSSION

To the best of our knowledge, this study is the first to examine the effects of a 12-week moderate-level oral dose of vitamin D₃ (4,000 IU) on acute and persistent changes in the fecal microbiota and metabolome of healthy adults. Overall, we found that a moderate dose of vitamin D₃ supplementation leads to significant changes in serum 25(OH)D levels after 12 weeks of intervention. With regard to changes in the fecal microbiota, we identified significant differences in β-diversity within the treatment group that led us to examine stability. Intriguingly, we observed an association between microbial stability and serum vitamin 25(OH)D levels. Specifically, there was a strong negative association between the percent change in serum 25(OH)D and microbiota stability. In addition, we found that vitamin D₃ supplementation altered specific taxa in the fecal microbiota both acutely (enriched *Bifidobacterium*, *Anaerostipes*, and diminished *Faecalibacterium*) and persistently (enriched *Erysipelotrichaceae* and diminished *Eubacterium coprostanoligenes* and *Prevotella*). Furthermore, we identified significant changes in microbial metabolites, mainly amino acids, and their networks resulting from vitamin D supplementation. These results indicate that a moderate dose of vitamin D₃ over 12 weeks is sufficient to significantly alter the fecal microbiota and metabolome in healthy adults leading to persistent changes in individual taxa and overall microbial stability.

Consistent with the results of previous research, our study demonstrated that a 12-week vitamin D₃ supplementation of 4,000 IU per day significantly increased serum 25(OH)D levels of the participants in the treatment group when compared to the placebo group. Importantly, those in the treatment group who were either clinically deficient

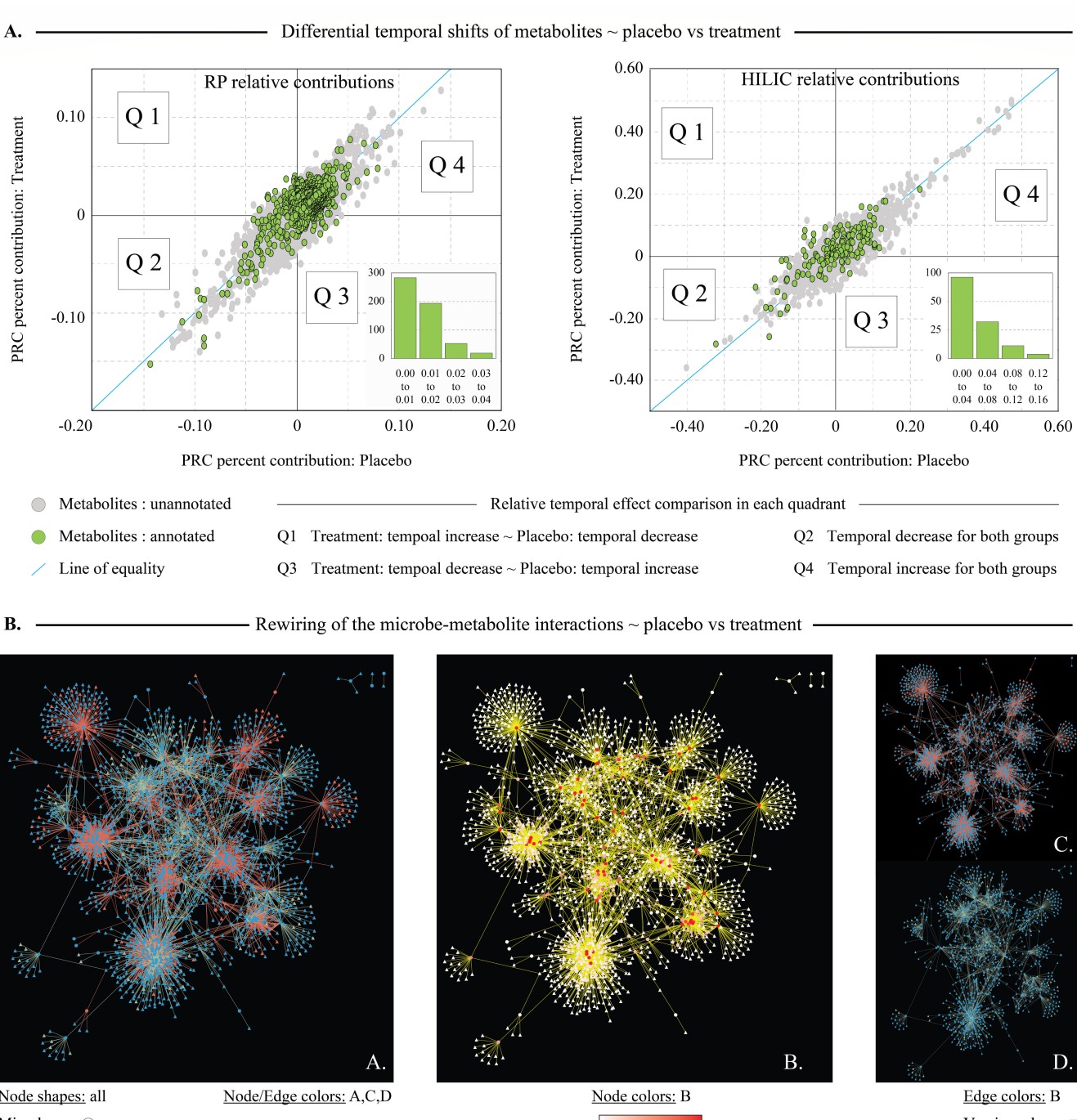

**FIG 6** Vitamin D$_3$ induces temporal changes of individual metabolites across placebo and treatment groups. (A) PRC was used separately to obtain the degree of contribution of each metabolite in both groups toward the overall change. These were standardized for each group and cross-compared by plotting the percent contribution of each metabolite obtained for PRC model, across placebo (x-axis: percentage) and treatment (y-axis: percentage) groups. Metabolites identified through RP and HILIC columns were analyzed separately and presented in respective plots. Each circle represents an individual metabolite, while green color circles represent annotated metabolites. Line of equality is used as the baseline for no change between two groups. The perpendicular distance from the central line depicts the degree of relative variation for each metabolite and the sample positioning attests to the directionality of the change, as per the quadrants highlighted. Each subplot shows the histogram of absolute perpendicular distances of annotated metabolites where the largest two bins were used for subsequent pathway explorations. (B) Differential microbe-metabolite interactions between placebo and the treatment groups. In each panel, circles represent microbes, triangles represent metabolites, and a connecting line attests to the interaction of Pearson's correlation of 75% or more between the

Fig 6 (Continued)

connected microbe and metabolite. Panels B_A and B_B show the DyNet rewiring between top correlated microbe-metabolite networks of individual placebo and treatment groups, which are shown in Panels B_C and B_D respectively. In panels B_A, B_C, and B_D, nodes/edges are colored based on their unique and shared nature as depicted in the figure legend. Panel B_A colors the rewired network based on the origin of the network, which shows that the majority of the nodes have been shared (86%: blue), while only a small number of connections are common between two individual networks (0.17%: gray). Panel B_B recolors the same DyNet rewiring, based on the degree of node rewiring, which further highlights the limited number of intact edges between two networks. This observation highlights that while these top correlated members in each group are mostly the same (86%), their interactions are overwhelmingly different between the groups (99.8%, panel B_B), suggesting differential metabolic programming due to vitamin D treatment.

(<20 ng/mL) or insufficient (<30 ng/mL) all became sufficient by 12 weeks. This is of clinical significance since large epidemiological studies report that 20% of adolescents and 29% of young adults in the USA have low levels (25–50 nmol/L) of serum 25(OH)D (39). Furthermore, evidence also supports a CRC risk and mortality reduction role of vitamin D if serum concentration of 25(OH)D is maintained above 10 ng/mL (40, 41), especially in early-onset CRC (42). Therefore, adequate 25(OH)D levels may serve as an important determinant for disease prevention and survival, especially in CRC. In addition, 25(OH)D deficiency has been recognized as a risk factor in inflammatory bowel conditions, including ulcerative colitis and Crohn's disease, which are both risk factors for CRC (43, 44). A meta-analysis investigating the association of serum 25(OH)D levels with CRC risk suggested that intake of 1,000–2,000 IU per day of vitamin $D_3$ could reduce CRC risk by 50% with serum concentration levels of 25(OH)D above 33 ng/mL (45). Likewise, vitamin $D_3$ supplementation may reduce the risk and mortality from CRC as well as early-onset CRC (5, 46–49).

By comparing microbial communities across different interventions, β-diversity helps in understanding how dietary components like vitamin $D_3$ supplementation influence microbiota composition. Our results demonstrated that microbial community composition was significantly altered by vitamin $D_3$ supplementation. While there was a significant change over time in β-diversity (Aitchison's distance) in the placebo group, intriguingly, we observed a significant increase in composition similarity in the treatment group, such that the samples appeared to cluster together across time. This suggests a potential effect of vitamin D on the gut microbiota in creating a more similar community structure. However, it is unclear if this is a direct effect on the microbiota or an indirect effect through feedback from the intestinal microenvironment. One possible explanation for the increased similarity of community structure among treatment groups could be resource availability (moderate vitamin $D_3$ dose) which may be driving growth and extinction events or blooms within the gut microenvironment in accordance with Gause's law (50, 51). However, much more research is required to understand the underlying mechanisms of how vitamin $D_3$ may be metabolized by specific microbes and induce changes in community structure over time, an area of active research in our group.

Additionally, after the 12-week intervention, we found significant shifts in specific microbial taxa in the fecal microbiota of individuals in the treatment group that were either acute or persistent. Specifically, there was a significant increase in *Bifidobacterium* after 2 weeks of daily vitamin $D_3$ supplementation among the treatment group. This increase in *Bifidobacterium* continued in the subsequent weeks of the study but was not significantly different from placebo by the end of the study. Similar findings of increased *Bifidobacterium* in response to vitamin $D_3$ supplementation have been noted in previous studies (16, 52, 53). *Bifidobacterium* spp., in general, show protective effects against disease and gastrointestinal disorders by mechanisms including secretion of antimicrobial peptides and exopolysaccharides (52, 54, 55). Huang et al. identified five protein-coding genes in *Bifidobacterium longum* LTBL16 correlating with a strong antioxidant activity (56). These genes were associated with oxygen tolerance and free radical scavenging via gene functions including, alkyl hydroperoxide reductase (subunit C), peroxiredoxin Q/bacterioferritin co-migratory protein (BCP), NADPH, NADH oxidase, and NAD-dependent protein deacetylase of the SIR2 family (56). Conversely,

a diminished population of *Bifidobacterium* has been recognized in individuals with disease states, including inflammatory bowel disease and cancer (57–63). Specifically, in a study comparing microbiota of healthy individuals to those with CRC, *Bifidobacterium* was found to be significantly diminished in those with CRC (64). In support of this finding, CRC patients in the same study also appeared to have lower concentrations of short-chain fatty acids, which are known to improve gut barrier function and exhibit anti-inflammatory, antimicrobial, and antitumorigenic activities (64, 65). *Bifidobacterium* was found to be protective against CRC tumor development in murine models of CRC through the downregulation of epidermal growth factor receptor (EGFR), human epidermal growth factor receptor 2 (HER-2), and prostaglandin-endoperoxide synthase 2, also known as cyclooxygenase-2 (PTGS-2 COX-2) (66, 67). Additionally, *Bifidobacterium* demonstrates an effect on the apoptotic resistance of cancer cells through activation of pro-caspases, upregulation of Bax proteins, and downregulation of anti-apoptotic Bcl-2, although exact mechanisms are poorly understood (68). In support, supplementation with 2 g of *Bifidobacterium longum* BB536 (powder) daily for 7–14 days preoperatively, with milk or water, among patients undergoing CRC surgery showed improved postoperative inflammatory response, promoted healthy recovery, and reduced the duration of hospital stay (69). Additionally, *Anaerostipes* and *Erysipelotrichaceae UCG-003* also gradually increased over time, both of which have been reported to be significantly reduced in CRC patients (70, 71). Conversely, we observed *Bacteroides* as the foremost decreasing genera in the vitamin D group, followed by *Prevotella* and *Faecalibacterium*. Both *Bacteroides* and *Prevotella* have previously been reported to decrease with vitamin D supplementation, while their levels have been observed to increase in CRC patients (72–74). Interestingly, we observed a decrease over time in *Faecalibacterium*, a prominent anti-inflammatory microbe in the gut within the vitamin D group, while previous studies have observed the relative abundance of this genus to potentially increase in similar conditions (75). Among these top temporally affected species, the influence of vitamin D on *Bifidobacterium, Anaerostipes, Faecalibacterium,* and *Bacteroides* exhibited a recovering trend by day 78 (64 days after last vitamin D administration). Conversely, *Prevotella*, *Erysipelotrichaceae UCG-003*, *Eubacterium coprostanoligenes,* and an unclassified *Firmicutes* demonstrated a more persistent response. Taken together with these studies, our results suggest that supplementation with vitamin $D_3$ among healthy individuals can alter the microbiota favorably, specifically increasing *Bifidobacterium* abundance, which has demonstrated beneficial effects in controlling inflammation, regulating proliferation, and potentially improving CRC outcomes.

We also used the KEGG reference genome to map these metabolites which revealed pathways such as amino acid metabolism, biosynthesis of secondary metabolites, microbial metabolism in diverse environments, mineral absorption, and ABC transporter pathways to be linked with the origin for these metabolites. In addition, the role of vitamin $D_3$ in stimulating amino acid uptake has already been observed in other animals, and amino acid pathways such as glycine, serine, and threonine metabolism have already been predicted to be significantly enriched upon vitamin $D_3$ supplementation (76, 77). Hence, targeted approaches, metagenomics, and metatranscriptomics are warranted to assess specific metabolic pathways of vitamin $D_3$ metabolism by specific microbial taxa in human gut. Collectively, this study offers preliminary evidence of metabolomic shifts resulting from vitamin D supplementation, which will require validation to confirm the origins and outcomes of these key metabolites.

Importantly, we present a novel exploration into the global changes in the human gut metabolome associated with vitamin D supplementation, leveraging untargeted metabolomics, an unprecedented endeavor, to our knowledge. Our findings reveal distinct temporal shifts induced by supplemental vitamin D, contrasting notably with the placebo metabolomes. Specifically, metabolites including methionine, acetylcholine, tryptophan, glutamine, and uric acid demonstrated significant temporal increases in abundance upon vitamin D supplementation. The essential amino acid tryptophan has been linked with CRC where the increased plasma levels have been associated with

lower risk of CRC development, while tryptophan receptor activation in the colon has been linked with tumor progression in CRC (78, 79). Specifically, in a mouse model of intestinal *Vdr* knockout, fecal metabolites were also significantly altered, including kynurenine, a tryptophan metabolite, and N-acetylglutamate (80). Similarly, methionine also has been observed to have a complex role in CRC, especially with some studies suggesting their restriction as a cancer prevention strategy while other studies observe positive effects of methionine intake against CRC progression (81–83). Acetylcholine, on the other hand, has been theorized to be a signaling molecule in CRC progression, despite numerous benefits including the regulation of intestinal motility and secretion and enteric neurotransmission (84–87). Similarly, the role of glutamine in CRC progression has been observed to be complicated, where certain studies have shown their positive effects in CRC prevention while others have suggested their possible role in CRC progression (88, 89). Interestingly, all of these metabolites have been known to be produced by human host-associated *Bifidobacterium*, which we also observed to be enriched in our treatment group (90–95). Noteworthy metabolites associated with vitamin D supplementation included uric acid (previously documented), alongside testosterone sulfate and indoxyl sulfate (96–98), all of which have been associated with modulating CRC development and progression. In addition, the role of vitamin $D_3$ in stimulating amino acid uptake has already been observed in other animals, and amino acid pathways such as glycine, serine, and threonine metabolism have been predicted to be significantly enriched upon vitamin $D_3$ supplementation (76, 77). Furthermore, given amino acid metabolism is significantly altered in colorectal tumors, including tryptophan and glutamine, identifying key microbial metabolic alterations from vitamin D supplementation will be critical (99–102). Thus, further studies are warranted to understand the specific links between vitamin $D_3$ supplementation and gut microbial modulation of these compounds by specific member in the communities. Lastly, it will also be critical to narrow down these individual metabolic signatures to precisely decipher how supplemental vitamin D interacts within gut microbiota from individuals with CRC compared to healthy, an area our labs are actively addressing. Overall, our study sheds light on shifts in microbe-metabolite interactions induced by vitamin D supplementation. Although the constituents of these interactions remained relatively consistent, we observed a notable alteration in microbe-metabolite interactions between treatment groups. This observation challenges the notion that vitamin D metabolism pathways are strictly confined to specific bacterial groups. While direct observations will be necessary, this finding supports the investigation of vitamin D metabolism by the gut microbiome, especially given the recent identification of cholesterol—a structurally similar nutrient— metabolism by specific gut microbiota (103, 104).

Despite favorable results, our study has some limitations. We enrolled a convenience sample of mostly college-aged students from our university, limiting the applicability of the results to a wider population. There were also differences in sugars (tapioca vs rice syrup) between the two formulations of intervention and placebo gummies, though this likely did not impact the microbiome significantly since this difference was not attributed to microbiota-accessible carbohydrates. While we were able to randomize our participants based on initial baseline serum 25(OH)D levels, we were not able to further randomize based on baseline microbiota composition, which led to significant differences in composition between groups at baseline. Microbiota analysis of all time points (instead of only four time points), using a higher resolution sequencing approach, may have provided more clinical significance into daily microbial compositional changes in response to vitamin $D_3$ supplementation; these studies are underway. We also could have included additional weekly samples to further improve our study design. Additionally, our microbiota analysis could have been impacted due to the collection of fecal samples and not mucosa-associated microbiota, which may not be a true representation of the microbiota in upper gastrointestinal tract, or the collection of fecal swabs instead of whole stool samples, which may not be representative of the entire sample. Furthermore, we were limited in this study to only 16S rRNA

sequencing as opposed to more comprehensive metagenomic/metatranscriptomic (e.g., Whole Genome Sequencing or RNA-seq) and metabolomic analyses. Additionally, we encountered common challenges inherent in untargeted metabolomics, notably the restricted depth of available metabolite annotations. However, efforts are underway to conduct these analyses with additional samples from our current study.

## CONCLUSION

Overall, our study extends the field of microbiome research in several ways. First, we analyzed the effects of a moderate dose of vitamin $D_3$ on the fecal microbiota composition, stability, and metabolites. Second, we were able to obtain serially collected samples at multiple time points, which improves our understanding of temporal variability within the microbial ecosystem in response to vitamin D. Third, we recruited healthy 25(OH)D sufficient adults to understand the effects of vitamin $D_3$ on the microbiota in the absence of a frank clinical deficiency, in opposition to those studies of disease states, which allowed us to understand potential disease prevention effects of vitamin $D_3$. In conclusion, the data from our study, and studies conducted previously, suggest a possible role of vitamin $D_3$ in modulating the gut microbiota, favorably by increasing the population of health-promoting bacteria and reducing the population of opportunistic pathogens. Metabolomics analysis also supported the vitamin D modulation by human gut microbiota, which was further asserted by the drastic rewiring observed among the top microbe-metabolite associations. Thus, our study furthers the understanding of other possible mechanisms by which vitamin D is acting to protect against CRC. This study supports further elucidating our mechanistic understanding of how vitamin D impacts the gut microbiome directly and indirectly through interactions with the host.

## ACKNOWLEDGMENTS

We thank all the participants and volunteers who helped in the Vitamin D Microbiome Trial. We also thank Dr. Abagail Johnson and Dr. Susan Williams for reviewing the manuscript. We also thank the University of Arizona Mass Spectrometry Core for their analytical chemistry expertise and data acquisition via LC-MS/ MS.

This research was funded by The Baylor Fellows Award, Baylor University, awarded to L.G. This research was supported, in part, by the Department of Energy, Office of Science Biological and Environmental Research Grant DE-SC0021349, awarded to M.M.T.

Conceptualization: M.W., A.C., and L.G.; methodology: M.W., A.C., M.M.T., M.Z., and L.G.; software: A.C., M.Z., and S.R.; data collection: M.W., A.C., J.L.H., J.S.F., and M.Z.; formal analysis: M.W., A.C., J.L.H., M.M.T., M.Z., S.R., and L.G.; data curation: A.C., M.W., M.Z., and S.R.; writing—original draft preparation: M.W. and S.R.; writing—review and editing: M.W., A.C., M.M.T., S.R., and L.G.; visualization: A.C., M.Z., and S.R.; supervision: L.G. and M.M.T.; project administration: M.W., L.G., and M.M.T.; funding acquisition, L.G.

## AUTHOR AFFILIATIONS

[1]Human Health Performance and Recreation, Robbins College of Health and Human Sciences, Baylor University, Waco, Texas, USA

[2]Human Science and Design, Robbins College of Health and Human Sciences, Baylor University, Waco, Texas, USA

[3]Nutrition Services Division, Walter Reed National Military Medical Center, Bethesda, Maryland, USA

[4]Department of Biology, Baylor University, Waco, Texas, USA

[5]Department of Environmental Science, University of Arizona, Tucson, Arizona, USA

[6]BIO5 Institute, The University of Arizona, Tucson, Arizona, USA

[7]Colorado Program for Musculoskeletal Research, Department of Orthopedics, University of Colorado Anschutz Medical Campus, Aurora, Colorado, USA

## AUTHOR ORCIDs

Madhur Wyatt http://orcid.org/0009-0000-7007-365X
Leigh Greathouse http://orcid.org/0000-0002-6855-8516

## FUNDING

| Funder | Grant(s) | Author(s) |
| --- | --- | --- |
| U.S. Department of Energy (DOE) | DE-SC0021349 | Malak M. Tfaily |
| Baylor University (BU) | N/A | Leigh Greathouse |

## AUTHOR CONTRIBUTIONS

Madhur Wyatt, Conceptualization, Data curation, Formal analysis, Investigation, Methodology, Project administration, Resources, Supervision, Validation, Visualization, Writing – original draft, Writing – review and editing | Gabriella Von Dohlen, Data curation, Formal analysis, Investigation, Methodology, Software, Visualization | Jeffery L. Heileson, Formal analysis, Investigation, Methodology, Writing – review and editing | Jeffrey S. Forsse, Investigation, Methodology, Project administration, Supervision, Writing – review and editing | Sumudu Rajakaruna, Data curation, Formal analysis, Investigation, Methodology, Software, Validation, Visualization, Writing – original draft, Writing – review and editing | Manja Zec, Data curation, Formal analysis, Investigation, Methodology, Software, Visualization | Malak M. Tfaily, Conceptualization, Data curation, Formal analysis, Funding acquisition, Investigation, Methodology, Project administration, Resources, Software, Supervision, Writing – review and editing | Leigh Greathouse, Conceptualization, Data curation, Formal analysis, Funding acquisition, Investigation, Methodology, Project administration, Resources, Supervision, Validation, Visualization, Writing – original draft, Writing – review and editing.

## DATA AVAILABILITY

All de-identified data and code used to produce these analyses are available at the following link: https://github.com/GreathouseLab/Vitamin_D_RCT
A list of raw reads, filtered reads, reads after filter, denoised reads, non-chimeric reads, and reads after rarefaction. SRA accession ID: PRJNA1051625.
Any de-identified clinical metadata or experimental data generated as part of this study will be made available upon reasonable request by the authors.

## ETHICS APPROVAL

The study was approved by the Institutional Review Board (IRB) (1845028-2) at Baylor University and registered under the identifier NCT05387876 at ClinicalTrials.gov. An informed written consent was obtained from each participant prior to study participation.

## ADDITIONAL FILES

The following material is available online.

### Supplemental Material

**Supplemental figures and tables (Spectrum00083-24-s0001.pdf).** Fig. S1-S11; Tables S1-S4.
**Supplemental material (Spectrum00083-24-s0002.xlsx).** Supplemental dataset.

### Open Peer Review

**PEER REVIEW HISTORY (review-history.pdf).** An accounting of the reviewer comments and feedback.

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
