## [Reviewer comments · Microbiology Spectrum]

Microbiology Spectrum

Randomized control trial of moderate dose vitamin D alters microbiota stability and metabolite networks in healthy adults

Madhur Wyatt, Ankan Choudhury, Gabriella Von Dohlen, Jeffery Heilesen, Jeffrey Forsse, Sumudu Rajakaruna, Manja Zec, Malak Tfaily, and Leigh Greathouse

Corresponding Author(s): Leigh Greathouse, Baylor University

Review Timeline:

Submission Date:	January 10, 2024
Editorial Decision:	February 26, 2024
Revision Received:	May 3, 2024
Editorial Decision:	June 5, 2024
Revision Received:	July 3, 2024
Accepted:	July 10, 2024

Editor: Jennifer Auchtung

Reviewer(s): Disclosure of reviewer identity is with reference to reviewer comments included in decision letter(s). The following individuals involved in review of your submission have agreed to reveal their identity: Funmilola Abidemi Ayeni (Reviewer #1)

Transaction Report:

DOI: <https://doi.org/10.1128/spectrum.00083-24>

Re: Spectrum00083-24 (Randomized control trial of moderate dose vitamin D alters microbiome stability and metabolite networks in healthy adults)

Dear Dr. Leigh Greathouse:

Thank you for the privilege of reviewing your work. Below you will find my comments, instructions from the Spectrum editorial office, and the reviewer comments.

Revision Guidelines

Sincerely,
Jennifer Auchtung
Editor
Microbiology Spectrum

Reviewer #1 (Comments for the Author):

The study reported several interesting observations. These include: specific serum 25(OH)D levels impact microbiome stability up to 50%, a significant shifts in specific microbial taxa in the gut microbiome of individuals after 12 weeks of vitamin D supplementation, a significant increase in Bifidobacterium after two weeks of intervention. vitamin D3 supplementation resulted in notable metabolic shifts. Some comments that I would have stated have been adequately addressed by the authors in the

limitation of the study.

The authors could have a full description of acronyms when first used e.g. CRC, (25(OH)D)

The authors could explain in the method why there is no midpoint or fecal sample collection from day 15 till day 77.

It would make the reading easier to navigate if processing of fecal material was described first before DNA extraction and other sections. It is currently described under Liquid Chromatography section. I believe the fecal samples from the swabs were initially processed before DNA extraction and LC/MS

Reviewer #2 (Comments for the Author):

The work discusses the response of the gut microbiome and metabolome upon daily supplementation of vitamin D of healthy individuals. An intervention study has been conducted and the individuals were sampled during the course of the intervention and post intervention. Amplicon sequencing and non-targeted metabolomics were used to study the microbiome and metabolome, and the vitamin D levels in the serum were measured. The authors explored microbial community changes due to intervention and tried to identify shifts of specific bacterial taxa. Correlations between bacterial taxa abundance and abundance of metabolites was presented in networks.

Below you can find some questions I have regarding the study. The questions are separated in the larger or more general questions, and smaller questions.

General questions:

Question 1: A lot of emphasis has been given in the abstract and introduction on colorectal cancer and how vitamin D can be used to prevent disease. In this study, the dietary intervention with vitamin D was conducted on healthy individuals and you investigated their gut microbial composition. I am wondering at this moment why you are focusing on the gut microbiome and if you are hypothesizing that the vitamin D effect on colon health is through the resident microbiota. Is there any previous knowledge that could support such notion? Information provided in introduction does not make a strong case for the connection of vitamin D's effect colorectal cancer through the microbiome. I find it very crucial to define in your introduction why you are considering important to look at changes in the microbiota and how this can contribute to support your statement on the beneficial effect of vitamin D at CRC, especially given that your study is conducted in healthy individuals and not CRC patients.

Question 2: As individuals participating in the study are healthy, how are their levels of vitamin D compared to suggested normal values before and after the intervention? You are registering a high increase in vitamin D for some individuals, is it still in the health limits? Can there be detrimental effects due to vitamin D oversupplementation?

Question 3(line 356): You base a lot the support of your hypothesis in the notion that gut microbiota stability is per-se a positive characteristic. It is only discussed for the first time in the discussion, in line 548, that it is not clear from this study if stability is beneficial for the host. You must consider that your definition of stability here is relative to the diversity observed in the placebo group (more stable than the placebo). However, it has been already shown that there are temporal changes in the gut microbiota of healthy individuals and these shifts are not indicative of any disease development. Thus, if these temporal shifts are not problematic, stability is not a direct indication of a 'healthier' gut microbiota. I would rather suggest that as the supplementation of vitamin D does not result in a microbiota shift, there is a minimal effect of the intervention in the general microbiota composition.

Question 4 (Line 370-385): How to three subgroups for each group were decided? They do not include a similar number of individuals, so how did you decide for the split? It also needs to be better explained why the subgroups are different between treatment and placebo group.

In addition, have you accounted for baseline differences among individuals in the different groups? In another point you are mentioning that some individuals are vitamin D deficient while others were not.

Finally, the type of analysis (Figure 3D) has some serious flaws or you have not explained properly what you performed. Are you using each timepoint comparison for a certain individual in the plot without accounting/controlling for the time? Your regressions do not look like they are fitting properly your data. I would suggest you consider another type of analysis for this part. You need to also reconsider if subgrouping your treatment dataset in three with such a low number of individuals provides you with enough statistical power. You also need to account for time.

Question (line 397): How did you test for the differentially abundant taxa? It is not described neither in M&M nor here. It is important to consider the shortcomings of 16S rRNA sequencing data and use an appropriate method for this analysis. Multiple papers are available that discuss the different existing methods and their benefits/limitations.

Question 5 (Line 409- 416): There are better methods to evaluate whether the abundance of certain taxon is correlated with β -diversity. In addition, I fail to see how the connection between the β -diversity change and the abundance change of certain taxa

supports what you are trying to argue here. Higher β -diversity between timepoints on its own does not indicate a certain health profile. It is probably more valuable to investigate the association between the change in the vitamin levels and the abundance of the specific taxa.

Question 6 (line 443 and Figure 5): Splitting your dataset at 50% vitamin D change at endpoint is creating two very unequal size groups and as at the same time you are not accounting for the sampling point or the initial vitamin D levels. In this part, your analysis has many shortcomings. It is also a very small dataset to train a Random Forest and you need to make sure you are not using the same individual in the training and testing subset, which I guess it is hard to avoid due to the dataset size. Overall, I find quite hard to use this type of analysis given the characteristics of your datasets. At the same time, it does not contribute with significant new information compared to the previous section and, thus, you can maybe skip it all together.

Question 7 (line 490): You have a nice dataset of both microbiota and metabolomic data. The metabolomic data especially can help you better understand potential mechanism behind the changes induced by the vitamin D supplementation. But the integration of the microbiota-metabolomics datasets is currently quite restricted.

Have you investigated with which metabolites are the previously identified taxa correlated? Specifically, are any of the bacterial taxa that you have also linked to CRC correlate with specific metabolites? In addition, there is no link between the metabolites the case you are trying to build with vitamin D protection against CRC. Is there any previous knowledge of molecules found in the colon linked with CRC?

Question 8: In discussion you are referring to SCFA as the potential mechanism through which the detected upregulated taxa might have a positive effect on the host. Where you able to detect any of those compounds in your metabolomics data?

Smaller questions

Line 26: Similarity of what?

Line 93: moderate dose - clarify how often this dose is received. Daily?

Figure 2: It is a rather peculiar way to plot the individual points below the barplot. Consider printing the points on the barplot instead.

Line 350: Why get an increase in placebo and not treatment? What kind of test was performed? Was it paired?

Line 359: Do you mean significant differences between the two groups across timepoints?

Figure 3C: There is little added value in this figure as all the samples are already displayed in figure 3B and this figure does not visually help to support the argument made in the text. Other ways instead of a PCoA (for example a barplot) should be considered to display the higher similarity across timepoints of the treated versus the placebo group.

Line 372-375: The sentence is hard to read. Please rephrase.

Line 418: A lot of information provided in this paragraph would better fit in the discussion rather than results section.

Figure 4A: It is very hard to distinguish differences in this color scale, especially the green color is almost impossible to differentiate from the blue.

Line 552-558: This part reads as a repetition of what has been already mentioned in the previous paragraph.

Line 558-569: It is a very important part of the overall hypothesis of the beneficial effect of vitamin D whether it has a direct effect on the microbiota. Nevertheless, there is a lot of speculation and not enough evidence provided in these sentences.

Line 581-606: This extensive review of vitamin D dosing is not relevant in the discussion of your findings and does not help to support any of your hypothesis. I believe it is redundant.

Line 615-634: You could re-format this paragraph to be more concise.

Line 654: How do your result support the cross-feeding?

Reviewer #3 (Comments for the Author):

I enjoyed reviewing the manuscript by Wyatt et al., which evaluated in humans the effect of 4000 UI vitamin D supplementation on the fecal microbiota. While the findings would benefit the field, I do have some concerns that should be addressed.

- On lines 452-455, it is mentioned that baseline serum 25(OH)D levels were the best predictor of serum changes in 25(OH)D. Do baseline values for 25(OH)D also predict the stability findings and why different groupings link to differences in the microbiota? First, added to Figure 2, the authors should include additional information related to the change in serum 25(OH)D levels as a factor of baseline levels. Similar, for Figure 3D, it would be good to clarify if baseline levels link to changes in 25(OH)D when based on the same grouping.
- The two main avenues by which vitamin D would modulate the gut microbiota is by i) not being absorbed by the host in the small intestine or ii) by being absorbed and favorably influencing the colonic environment and the immune system. If the microbiota are using the vitamin D as a growth substrate, then one would expect less changes in serum 25(OH)D levels (which is less desirable) and gut microbiota changes. The authors should discuss these relationships as they related to the study findings.
- The study stated that data on body composition, dietary intake assessment, physical activity and sun exposure were collected. However, there was no discourse on these subjects to show if there were any relationship/association between them, serum vitamin D levels and microbiome changes. It would benefit the readers to include more information here, test whether any of these factors or potential covariates changed during the intervention, and also use these variable in the random forest analysis performed to determine if any host factors explain the change in Vitamin D.
- Lines 636 to 664. The data presented in this manuscript does not provide information related to changes in inflammation, CRC risk, or SCFAs, and it does not evaluate cross-feeding interactions with butyrate producers. For instance, a response to Bifidobacterium does not equal an anti-inflammatory effect in humans. These sentences seem to be overreaching based on the study findings and I would encourage the authors to remove them and focus the discussion on the findings presented.
- To ease burden to reader, I would recommend that the authors break-up some of their large paragraphs into smaller paragraphs. For examples lines 443 to 467 should be broken into 2 paragraphs, with a paragraph presenting regression findings and another discussing random forest. This is also the case for lines 571 to 638.
- I would encourage the authors references the vocabulary proposed by Marchesi and Ravel, 2015 (<https://pubmed.ncbi.nlm.nih.gov/26229597/>). For instance, throughout the manuscript there is reference to the microbiome; however, only 16S rRNA gene amplicon sequencing was used, which would assess the microbiota. There is reference as well to the distal gut microbiome, where in reality what is being assessed in the fecal microbiota, as a proxy for the gut microbiota and not specifically the distal gut microbiota.
- Please add to the methods information on how the randomization was performed.
- Please add more information about the treatments. Specifically the weight of product consumed for each treatment (i.e. gram/gummy) and how the gummies were provided, as in were daily bags or was a single container provided. How was the intervention blinded? How was protocol adherence evaluated? Was the change in serum 25(OH)D levels related to protocol adherence?
- Stool sample collection is not clear. Was this done by rectal swab or was a sample collected and then the fecal sample swabbed?
- It is unclear if OTUs or ASVs are being presented, although based on the pipeline presented ASVs would be assumed. Please add information on the sequence percent similarity used for generating OTUs or ASVs.
- There is inconsistencies on how the p-values are presented. Would recommend using $p < 0.0001$ for those values presented in the text with scientific notation.
- I would encourage the authors to proofread the manuscript again for grammar errors. For instance, line 67, 'was' should be 'were' and line 521, 'the' should be 'that'.
- Lines 426 to 435. Refrain from discussing the study findings related to the literature in the results.
- Lines 520 to 521. This information should be reported in the results and could even be even added to Figure 2.
- Lines 717. Term dose-dependent is used, however, this study design is not able to evaluate dose dependent effects.

- While different intervention and control gummies were used that differed in their carbohydrate sources (tapioca vs. rice syrup), it is very unlikely that these differences would alter the microbiota unless one of the syrups contained IMO or dietary fibers like resistant maltodextrin. However, the gummies do/may differ in other ingredients such as the carnauba wax and pectin. Therefore, the limitation would be that the control is not a match formulation without the vitamin D.
- Figure 5. Unclear in legend which timepoint is being used for the analysis.

Reviewer #1 (Comments for the Author):

The study reported several interesting observations. These include: specific serum 25(OH)D levels impact microbiome stability up to 50%, a significant shifts in specific microbial taxa in the gut microbiome of individuals after 12 weeks of vitamin D supplementation, a significant increase in Bifidobacterium after two weeks of intervention. vitamin D3 supplementation resulted in notable metabolic shifts. Some comments that I would have stated have been adequately addressed by the authors in the limitation of the study.

The authors could have a full description of acronyms when first used e.g. CRC, (25(OH)D)

Author Response: Thank you for allowing us to address this issue, it has been corrected.

The authors could explain in the method why there is no midpoint or fecal sample collection from day 15 till day 77.

Author Response: Thank you for allowing us to address this concern. We originally designed this project as a pilot project using the study design from Johnson et al. (<https://doi.org/10.1016/j.chom.2019.05.005>). Specifically, as this was funded as an undergraduate-engaged research project as part of an internal award, we used the study design by Johnson et al. as our guide. Unfortunately, limited funding led to decisions that in hindsight could have been better optimized, which as the reviewer points out astutely could have included a midpoint sample. In future studies, with ample funding, we would more likely include weekly samples and sequencing of all samples. We have now included this in the description of the limitations of the study in the discussion section.

It would make the reading easier to navigate if the processing of fecal material was described first before DNA extraction and other sections. It is currently described under Liquid Chromatography section. I believe the fecal samples from the swabs were initially processed before DNA extraction and LC/MS

Author Response: We have now added the following text to the section on DNA extraction from stool samples: "The brief procedure is as follows, the Sterile Flocked Collection Swab was detached from the stalk and added to the bead bashing tubes supplied, followed by physical disruption by bead beating (TissueLyzer II) at 30 Hz for 30 mins. Then the supernatant was separated by centrifugation at 3000 g for 5 minutes, lysed with proprietary genomic lysis buffer, ran through Silicon-based DNA filter and purified by multiple washing. The resultant ultra-pure DNA was used for 16s rRNA amplicon sequencing".

Reviewer #2 (Comments for the Author):

The work discusses the response of the gut microbiome and metabolome upon daily supplementation of vitamin D of healthy individuals. An intervention study has been conducted and the individuals were sampled during the course of the intervention and post intervention. Amplicon sequencing and non-targeted metabolomics were used to study the microbiome and metabolome, and the vitamin D levels in the serum were measured. The authors explored microbial community changes due to intervention and tried to identify shifts of specific bacterial taxa. Correlations between bacterial taxa abundance and abundance of metabolites was presented in networks.

Below you can find some questions I have regarding the study. The questions are separated in the larger or more general questions, and smaller questions.

General questions:

Question 1: A lot of emphasis has been given in the abstract and introduction on colorectal cancer and how vitamin D can be used to prevent disease. In this study, the dietary intervention with vitamin D was conducted on healthy individuals and you investigated their gut microbial composition. I am wondering at this moment why you are focusing on the gut microbiome and if you are hypothesizing that the vitamin D effect on colon health is through the resident microbiota. Is there any previous knowledge that could support such notion? Information provided in introduction does not make a strong case for the connection of vitamin D's effect colorectal cancer through the microbiome. I find it very crucial to define in your introduction why you are considering important to look at changes in the microbiota and how this can contribute to support your statement on the beneficial effect of vitamin D at CRC, especially given that your study is conducted in healthy individuals and not CRC patients.

Author Response: Thank you for allowing us to define this hypothesis further, if we understand the reviewer's question correctly, regarding the connection between vitamin D, the gut microbiota/microbiome, and vitamin D. First, significant evidence from both animal and human studies suggests that the gut microbiome plays a crucial role in the development of colorectal cancer (CRC) [1][2][3]. In a healthy state, the gut microbiome is dominated by diverse obligate anaerobes. However, under host inflammatory pressure, the gut microbiome can shift to a state of reduced diversity, which may be dominated by facultative anaerobes, including pathogenic species such as Bacteroides fragilis, Escherichia coli pks+, and Fusobacterium nucleatum[4][5]. Animal studies have demonstrated that several of these pathogenic bacteria can promote inflammation, colitis, and intestinal carcinogenesis. One dietary nutrient that has shown the ability to combat inflammation, reduce colitis, and potentially decrease the incidence of CRC is vitamin D [6]. Vitamin D can regulate gut permeability, induce the secretion of antimicrobial peptides, and aid in the detoxification of the secondary bile acid lithocholic acid. In human studies, clinical trials have shown that certain probiotics can increase serum vitamin D levels [7]. Recently, a double-blind, Phase II, randomized trial involving 74 individuals diagnosed with CRC found that supplementation with 2,000 IU/day of vitamin D for one year significantly altered the gut microbiome [8]. Specifically, those with high levels of the pathogen F. nucleatum had lower disease-free survival, while those who reached sufficient serum vitamin D levels (>30 ng/mL) had significantly reduced levels of this pathogen. These findings suggest that the gut microbiome may mediate the relationship between vitamin D and colon carcinogenesis[1][2][3].

[1] Ramos-Molina, B., et al. (2020). The Role of the Gut Microbiome in Colorectal Cancer Development and Therapy Response. Cancers, 12(6), 1406.

[2] Cheng, Y., et al. (2020). The Intestinal Microbiota and Colorectal Cancer. Frontiers in Immunology, 11, 615056.

[3] Dzutsev, A., et al. (2017). The Role of the Microbiota in Inflammation, Carcinogenesis, and Cancer Therapy. European Journal of Immunology, 47(2), 224-235.

[4] Feng, Q., et al. (2015). Gut Microbiome Development Along the Colorectal Adenoma-Carcinoma Sequence. Nature Communications, 6, 6528.

[5] Nakatsu, G., et al. (2015). Gut Mucosal Microbiome Across Stages of Colorectal Carcinogenesis. *Nature Communications*, 6, 8727.

[6] Kim H, et al. (2021). Total Vitamin D Intake and Risks of Early-Onset Colorectal Cancer and Precursors. *Gastroenterology*. 2021 Oct;161(4):1208-1217.

[7] Abboud M, et al. (2020). The Health Effects of Vitamin D and Probiotic Co-Supplementation: A Systematic Review of Randomized Controlled Trials. *Nutrients*. 2020 Dec 30;13(1):111.

[8] Bellerba F, et al. (2022). Colorectal cancer, Vitamin D and microbiota: A double-blind Phase II randomized trial (ColoViD) in colorectal cancer patients. *Neoplasia*. 2022 Dec;34:100842.

Question 2: As individuals participating in the study are healthy, how are their levels of vitamin D compared to suggested normal values before and after the intervention? You are registering a high increase in vitamin D for some individuals, is it still in the health limits? Can there be detrimental effects due to vitamin D over-supplementation?

Author Response: Thank you for allowing us to address these concerns. The US Endocrine Society has suggested defining cutoff values as follows: deficiency as serum 25-hydroxyvitamin D values ≤ 20 ng/mL (≤ 50 nmol/L), insufficiency as serum 25-hydroxyvitamin D values of 21 to 29 ng/mL (51 to 74 nmol/L), and sufficiency as serum 25-hydroxyvitamin D values of 30 to 100 ng/mL (<https://pi.oregonstate.edu/mic/vitamins/vitamin-D#deficiency>)

According to this cut-off, the following are the results from our dataset (below) showing that 2 participants are outside the sufficient limits. As soon as we confirmed they were above the limit, we notified them immediately and consulted them on how to reduce their serum vitamin D and to follow up with their physician. Furthermore, there are health problems that can present in people with long-term high intakes of vitamin D, which usually include hypercalcemia seen in older adults with other co-morbidities. DOI: 10.1016/j.jsbmb.2018.12.002

Serum 25(OH)D Before Intervention			Serum 25(OH)D After Intervention		
	Placebo	Treatment		Placebo	Treatment
Deficiency, ≤ 20 ng/mL	(n=1) 13.711	(n=1) 15.926	Deficiency, ≤ 20 ng/mL	(n=1) 17.341	0
Insufficiency, 21 to 29 ng/mL	(n=4) 25.662 27.671 21.4 22.578	(n=4) 26.295 28.874 23.574 21.522	Insufficiency, 21 to 29 ng/mL	(n=4) 28.617 24.367 28.394 27.815	0
Sufficiency, 30 to 100 ng/mL	(n=14) 39.644 62.101 49.757 49.12 58.835 31.044 47.626 40.311 30.12 43.87 53.423 35.126 51.588 41.784	(n=15) 30.922 30.08 58.221 40.208 43.702 60.006 47.81 44.722 41.358 34.9 39.097 52.463 51.256 76.895 49.183	Sufficiency, 30 to 100 ng/mL	(n=15) 36.308 34.682 42.514 44.869 43.696 62.484 31.61 38.601 38.959 43.814 30.1 48.481 32.157 35.77 38.565	(n=18) 70.151 52.475 75.695 61.912 49.058 74.913 72.351 45.404 51.717 84.571 64.77 73.446 52.619 46.748 85.229

					63.586 53.879 77.879
			Above 100 ng/mL		(n=2) 142.885 152.774

Question 3(line 356): You base a lot the support of your hypothesis in the notion that gut microbiota stability is per-se a positive characteristic. It is only discussed for the first time in the discussion, in line 548, that it is not clear from this study if stability is beneficial for the host. You must consider that your definition of stability here is relative to the diversity observed in the placebo group (more stable than the placebo). However, it has been already shown that there are temporal changes in the gut microbiota of healthy individuals and these shifts are not indicative of any disease development. Thus, if these temporal shifts are not problematic, stability is not a direct indication of a 'healthier' gut microbiota.

I would rather suggest that as the supplementation of vitamin D does not result in a microbiota shift, there is a minimal effect of the intervention in the general microbiota composition.

Author Response: Thank you for allowing us to clarify this issue. The reviewer is correct that we do not see any significant shift in alpha diversity. However, we do see a small but significant effect on composition between the two groups (Fig. 3B-C). What we intend to show, as the reviewer points out, is that the composition in the treatment group becomes more similar as the study progresses in the treatment group compared to the placebo group in the last two time points. To better show this, we have replaced our original figure with one that shows this direct comparison in composition.

It was not our intent to intimate that short-term gut microbiota stability is a positive characteristic. We recognize that stability, as measured relative to the placebo group, does not inherently equate to a 'healthier' gut microbiome, and we agree that temporal shifts in microbiota composition can occur in healthy individuals without indicating disease progression, as in response to drastic dietary changes, drug intake, or other stressors. Rather, we are indicating that stability can be a sign of resilience to external stressors – infection, antibiotics or other environmental stressors, which was demonstrated in the recent long-term cohort study (Fig. 3F) (<https://doi.org/10.1016/j.chom.2024.02.012>) comparing insulin resistant to insulin-sensitive individuals, not an indicator of disease development. We propose that vitamin D may be creating resiliency through direct and indirect interactions with the microbiota and host, which would require empirical testing in future in vivo studies.

In order to ensure that we are not causing the reader to make these conclusions we have amended our statements and conclusions to make more clear our interpretations.

Question 4 (Line 370-385): How to three subgroups for each group were decided? They do not include a similar number of individuals, so how did you decide for the split? It also needs to be better explained why the subgroups are different between treatment and placebo group. -

Author Response: After re-evaluating these data we agree with the reviewer that creating thresholds was not the appropriate approach due to our small sample size. Thus, we used all of the data to evaluate the linear relationship between changes in vitamin D and microbiota stability. This analysis demonstrated an inverse linear relationship between change in serum

vitamin D and microbiome stability, which was only significant in the treatment group. This is now represented in a new Fig. 3.

In addition, have you accounted for baseline differences among individuals in the different groups? In another point you are mentioning that some individuals are vitamin D deficient while others were not.

Author Response: Thank you for allowing us to clarify. To your first question, yes, we accounted for baseline differences between groups in two ways. First, we evaluated the key or potentially confounding variables (included in Table 1) and found that only baseline alpha diversity was significantly different between groups. Unfortunately, while we would have preferred to not only randomize based on baseline 25(OH)D levels, but also baseline gut microbiota composition, our resources did not allow for this at the time. The second way we accounted for baseline differences was in our random forest model, which took into consideration the relative abundance of the taxa most predictive of change in 25(OH)D levels. To your second point, we randomized all participants based on baseline 25(OH)D levels prior to assigning to placebo or treatment, so that each participant (except one) had a closely matching member in the other group. This is how we accounted or controlled for baseline serum 25(OH)D levels, along with levels of deficiency or insufficiency at baseline. There were only two participants who were clinically deficient, one in each group so we cannot infer any associations.

Finally, the type of analysis (Figure 3D) has some serious flaws or you have not explained properly what you performed. Are you using each timepoint comparison for a certain individual in the plot without accounting/controlling for the time? Your regressions do not look like they are fitting properly your data. I would suggest you consider another type of analysis for this part. You need to also reconsider if subgrouping your treatment dataset in three with such a low number of individuals provides you with enough statistical power. You also need to account for time.

Author Response: Thank you for pointing out this issue in our analysis. In Figure 3D the calculation of We agree that our data sample size does hold up to this type of threshold analysis, and thus we re-analyzed our data using a standard linear regression approach to fit the median of the inverse Atchison's distance for each participant to the percent change in serum vitamin D levels by group. Additionally, in the supplemental figures, we present a spline approach to fit these data (Figure S6). Furthermore, we consulted with our statistician, Dr. Jun Chen (Mayo Clinic), and he also agrees, that since we used the time information when we defined the stability using consecutive time points, the stability is a subject-level characteristic that is extracted using the four-time points. Thus, we believe our approach now is appropriate. We feel these approaches now more accurately represent the association between microbiota stability and the percent change in levels of serum vitamin D. The new figure is below and in the main text figures (Fig. 3D).

Question (line 397): How did you test for the differentially abundant taxa? It is not described neither in M&M nor here. It is important to consider the shortcomings of 16S rRNA sequencing data and use an appropriate method for this analysis. Multiple papers are available that discuss the different existing methods and their benefits/limitations.

Author Response: Thank you for allowing us to clarify. We very much appreciate the importance of choosing the appropriate test and correction testing factors associated with the nature of analyzing zero-inflated compositional data, like that of the gut microbiota. To answer your question, we chose to analyze the differential abundance of the 16S rRNA data using DESeq2. We have now added this to our material and methods sections.

Question 5 (Line 409- 416): There are better methods to evaluate whether the abundance of certain taxon is correlated with β -diversity. In addition, I fail to see how the connection between the β -diversity change and the abundance change of certain taxa supports what you are trying to argue here. Higher β -diversity between timepoints on its own does not indicate a certain health profile. It is probably more valuable to investigate the association between the change in the vitamin levels and the abundance of the specific taxa.

Author Response: Thank you for this point. We agree that a better analysis is performing Spearman correlation between change in center log ratio (CLR) taxon abundance and percent change in serum vitamin D. We have replaced Figure 4D with these new graphs to better illustrate individual taxa that significantly correlated with changes in serum vitamin D.

Question 6 (line 443 and Figure 5): Splitting your dataset at 50% vitamin D change at endpoint is creating two very unequal size groups and as at the same time you are not accounting for the sampling point or the initial vitamin D levels. In this part, your analysis has many shortcomings. It is also a very small dataset to train a Random Forest and you need to make sure you are not using the same individual in the training and testing subset, which I guess it is hard to avoid due to the dataset size. Overall, I find quite hard to use this type of analysis given the characteristics

of your datasets. At the same time, it does not contribute with significant new information compared to the previous section and, thus, you can maybe skip it all together. –

Author Response: We agree with the reviewer after reflection, that the small and unequal grouping make RF an inferior analysis choice, and have removed it from the main text figures.

Question 7 (line 490): You have a nice dataset of both microbiota and metabolomic data. The metabolomic data especially can help you better understand potential mechanism behind the changes induced by the vitamin D supplementation. But the integration of the microbiota-metabolomics datasets is currently quite restricted.

Have you investigated with which metabolites are the previously identified taxa correlated? Specifically, are any of the bacterial taxa that you have also linked to CRC correlate with specific metabolites? In addition, there is no link between the metabolites the case you are trying to build with vitamin D protection against CRC. Is there any previous knowledge of molecules found in the colon linked with CRC?

Author Response: We appreciate the reviewer's insightful comments, which prompted us to look deeper into the microbiota-metabolomics integration. As suggested, we have addressed each of the three points:

*1) **Key pathways for vitamin D metabolism:** while we identified significantly altered metabolites, pinpointing the exact microbe responsible for each metabolite change using current metabolomics techniques alone (without metatranscriptomics data) remains challenging. To best address this, we mapped these key metabolites to KEGG reference pathways (Supplementary Workbook 1), to identify key pathways that were significantly changing upon vitamin D supplementation. As you can see in the supplementary tables, these changes align with previous observations of gut microbiota metabolizing vitamin D3 through amino acid pathways. However, we acknowledge the limitations of using metabolomics alone, and future integration of metatranscriptomics data could provide a more definitive understanding of the specific microbial contributions to vitamin D metabolism.*

*2) **Correlations between metabolites and microbes:** we analyzed correlations between significantly changing metabolites and microbes. Our findings revealed moderately strong positive and negative correlations between these entities (Supplementary Workbook 1). This suggests direct evidence of interactions between these significantly changing microbe/metabolites, but also points to a complex interplay, possibly involving microbe-metabolite rewiring (as we have shown previously in Figure 6B). The observed correlations, while informative, highlight the need for further exploration using multi-omics approaches to unravel the precise nature and mechanisms underlying these microbe-metabolite relationships.*

*3) **CRC-associated metabolites and microbes:** we identified some microbes and metabolites with established links to CRC based on existing literature. A new section has been added to the manuscript explaining the complex roles of these elements in CRC development. Importantly, our analysis also revealed several novel associations between specific microbes, metabolites, and CRC that have not been previously reported. These findings provide additional insights into the complex gut microbiome-metabolome interactions that may contribute to colorectal cancer pathogenesis.*

Question 8: In discussion you are referring to SCFA as the potential mechanism through which the detected upregulated taxa might have a positive effect on the host. Were you able to detect any of those compounds in your metabolomics data?

Author Response: Thank you for allowing us to clarify. The discussion of SCFA was in relationship to the enrichment of *Bifidobacterium*, a SCFA producer, by vitamin D treatment. However, we have now amended this section and only note that this relationship exists so as to not overstate what our data shows.

Line 26: Similarity of what?

Author Response: We are referring to microbiota composition within the treatment group in response to supplemental vitamin D.

Line 93: moderate dose - clarify how often this dose is received. Daily?

Author Response: Thank you for allowing us to clarify. Yes, this is a daily dose of 4000 IUs of vitamin D3 for 12 weeks. We have added this additional sentence for clarification in the Study Design section.

Figure 2: It is a rather peculiar way to plot the individual points below the barplot. Consider printing the points on the barplot instead.

Author Response: We appreciate this comment. We attempted other methods of representing these data, however, we still feel this most accurately captures the interpretation of the results showing the details of the spread and the frequency of each group, and variability in response to vitamin D supplementation.

Line 350: Why get an increase in placebo and not treatment? What kind of test was performed? Was it paired?

Author Response: Due to significant heterogeneity in human gut microbiota it is possible that the placebo group had a unique fluctuation in alpha diversity not observed in the treatment group. It is possible that this lack of change in the treatment group could be due to the acute effects of vitamin D3 or just random chance. The test used for this analysis across time points was a Wilcoxon paired test and differences between days was Wilcoxon's non-paired test.

Line 359: Do you mean significant differences between the two groups across timepoints?

Author Response: Correct, in Fig. 3B this is comparing Aitchison's distance across timepoints between the two groups using the PERMANOVA test.

Figure 3C: There is little added value in this figure as all the samples are already displayed in figure 3B and this figure does not visually help to support the argument made in the text. Other ways instead of a PCoA (for example a barplot) should be considered to display the higher similarity across timepoints of the treated versus the placebo group.

Author Response: Thank you for this suggestion. We have now performed a PermDisp analysis in which the sample's distance from the centroid (i.e., dispersion) is compared between two

groups by PERMANOVA using `vegan's betadisper()` function using samples from days 7, 14, and 78. We believe this is a more statistically valid option than what was done previously. Both methods showed the same result that along with the mean of beta distances being different (shown by PERMANOVA in figure 3B), the beta dispersion is also significantly different. The new figure is below and Figure 3 in the manuscript.

Line 372-375: The sentence is hard to read. Please rephrase.

Author Response: Thank you, we have revised the sentence to reflect our new analyses. "While the participants in the placebo group had no correlation between stability and percent change in serum 25(OH)D, those in the treatment group demonstrated a significant inverse association between stability and increased serum 25(OH)D levels ($R=-0.052$, $R^2=-0.27$, $p<0.019$)."

Line 418: A lot of information provided in this paragraph would better fit in the discussion rather than results section.

Author Response: We have now removed that section and revised the discussion.

Figure 4A: It is very hard to distinguish differences in this color scale, especially the green color is almost impossible to differentiate from the blue.

Author Response: We have now updated this figure with blue and red colors instead.

Line 552-558: This part reads as a repetition of what has been already mentioned in the previous paragraph.

Author Response: We have now removed that section of the discussion since it did not convey any importance to the main findings.

Line 558-569: It is a very important part of the overall hypothesis of the beneficial effect of vitamin D whether it has a direct effect on the microbiota. Nevertheless, there is a lot of speculation and not enough evidence provided in these sentences.

Author Response: Thank you. We have now revised the discussion

Line 581-606: This extensive review of vitamin D dosing is not relevant in the discussion of your findings and does not help to support any of your hypothesis. I believe it is redundant.

Author Response: We have now removed that section of the discussion

Line 615-634: You could re-format this paragraph to be more concise.

Author Response: Thank you. We have now revised the discussion

Line 654: How do your result support the cross-feeding?

Author Response: We have now removed that section of the discussion

Reviewer #3 (Comments for the Author):

I enjoyed reviewing the manuscript by Wyatt et al., which evaluated in humans the effect of 4000 UI vitamin D supplementation on the fecal microbiota. While the findings would benefit the field, I do have some concerns that should be addressed.

On lines 452-455, it is mentioned that baseline serum 25(OH)D levels were the best predictor or serum changes in 25(OH)D. Do baseline values for 25(OH)D also predict the stability findings and why different groupings link to differences in the microbiota? First, added to Figure 2, the authors should include additional information related to the change in serum 25(OH)D levels as a factor of baseline levels. Similar, for Figure 3D, it would be good to clarify is baseline levels link to changes in 25(OH)D when based on the same grouping.

Author Response: We appreciate these insightful comments. Due to other comments from reviewers we have revised Fig. 3D to now show a linear relationship between stability change in baseline serum 25(OH)D.

The two main avenues by which vitamin D would modulate the gut microbiota is by i) not being absorbed by the host in the small intestine or ii) by being absorbed and favorably influencing the colonic environment and the immune system. If the microbiota are using the vitamin D as a growth substrate, then one would expect less changes in serum 25(OH)D levels (which is less desirable) and gut microbiota changes. The authors should discuss these relationships as they related to the study findings.

Author Response: We thank you for allowing us to clarify this important concept. We agree with the two main avenues presented by the review, but would also add a third - metabolism and activation of vitamin D into another metabolite of vitamin D or activation of vitamin D to 1,25 OH

D. The latter is, in fact, a known mechanism by which at least two species of bacteria metabolize vitamin D. Furthermore, given that cholesterol and vitamin D are very similar in structure, and certain bacteria metabolize cholesterol into coprostanol, it leads to the possibility that similar bacteria metabolize vitamin D into other vitamin D metabolites that are taken up and activated by renal system. This is currently under investigation in our lab.

The study stated that data on body composition, dietary intake assessment, physical activity and sun exposure were collected. However, there was no discourse on these subjects to show if there were any relationship/association between them, serum vitamin D levels and microbiome changes. It would benefit the readers to include more information here, test whether any of these factors or potential covariates changed during the intervention, and also use these variable in the random forest analysis performed to determine if any host factors explain the change in Vitamin D.

Author Response: Thank you for this comment. Initially, we conducted these analyses, however, they were not significant and as such did not show those data. However, we have now performed a correlation analysis between these variables and their association with serum vitamin D or the microbiome – now in supplemental figures

Lines 636 to 664. The data presented in this manuscript does not provide information related to changes in inflammation, CRC risk, or SCFAs, and it does not evaluate cross-feeding interactions with butyrate producers. For instance, a response is Bifidobacterium does not equal an anti-inflammatory effect in humans. These sentences seem to be overreaching based on the study findings and I would encourage the authors to remove them and focus the discussion on the findings presented.

Author Response: We have now removed that section of the discussion.

To ease burden to reader, I would recommend that the authors break-up some of their large paragraphs into smaller paragraphs. For examples lines 443 to 467 should be broken into 2 paragraphs, with a paragraph presenting regression findings and another discussing random forest. This is also the case for lines 571 to 638.

Author Response: Thank you. We have now revised the discussion

I would encourage the authors references the vocabulary proposed by Marchesi and Ravel, 2015 (

<https://pubmed.ncbi.nlm.nih.gov/26229597>

/). For instance, throughout the manuscript there is reference to the microbiome; however, only 16S rRNA gene amplicon sequencing was used, which would assess the microbiota. There is reference as well to the distal gut microbiome, where in reality what is being assessed in the fecal microbiota, as a proxy for the gut microbiota and not specifically the distal gut microbiota.

Author Response: Thank you. We appreciate these distinctions and have attempted to ensure that the most appropriate usage of microbiome vs microbiota is used throughout as well as the term fecal microbiota instead of distal gut microbiome.

Please add to the methods information on how the randomization was performed. –

Author Response: Thank you, we have added more details on randomization, which were also requested from another reviewer.

Please add more information about the treatments. Specifically the weight of product consumed for each treatment (i.e. gram/gummy) and how the gummies were provided, as in were daily bags or was a single container provided. How was the intervention blinded? How was protocol adherence evaluated? Was the change in serum 25(OH)D levels related to protocol adherence?

Author Response: Thank you. We have now added additional details on treatments below and to the manuscript:

- *Each chewable Vitamin D3 gummy provides 1000 IU (25 mcg) of cholecalciferol. Participants were required to consume 4 gummies per day, a total of 4000 IU*
- *The gram weight of each gummy is not mentioned on the packaging. See link: https://www.amazon.com/Nordic-Naturals-Vitamin-Gummies-Cholecalciferol/dp/B00G5APNBK/ref=sr_1_3_sspa?sr=8-3-spons&sp_csd=d2lkZ2V0TmFtZT1zcF9hdGY&psc=1*
- *The gummies were provided in a storage white pill bottle with a lid, bearing only the participant ID. No other items were listed on the bottle. Before distribution, participants were asked about any allergens they had to ensure they were not allergic to any ingredients in the gummies. (this is how the intervention was disguised).*
- *We provided weekly gummies (28 gummies per week) to each participant for the first 2 weeks, then we supplied the 3rd batch of gummies to the participants on day 14 for 33 days (132 gummies in 2 white pill bottles). The 4th batch of gummies was provided to the participants on day 49 for the remainder of the study (for 29 days; 116 gummies). At this point, the remainder of the gummies for the rest of the study were provided to the participants (see study design).*
- *We evaluated the adherence by asking participants weekly if they consumed gummies each day. We did this in person for the 1st 2 weeks. We then reached out to them via email to ensure compliance.*
- *Protocol adherence is listed below in the table.*

Compliance Rate			
	Expected	Received	Compliance %
Stool Samples	645	615	95%
ASA-24	645	572	87%
DHQIII Questionnaire	42	42	100%

Based on our evaluation of each participant during visits and emails, we believe compliance was 100%. Specifically, the research staff running the study spoke with participants personally when they met for visits to ensure they were consuming gummies regularly. Several of the participants brought to their notice that the gummies were so delicious that they were looking forward to consuming them. This gave us confidence we had 100% compliance.

The stool sample collection is not clear. Was this done by rectal swab or was a sample collected and then the fecal sample swabbed?

Author Response: We utilized the identical protocol for fecal sampling as is provided by the laboratory of Dr. Rob Knight - https://knightlab.ucsd.edu/wordpress/?page_id=478 – Specifically the “Detailed Instructions for Sampling”. We have added additional information to the Methods – Stool sections.

It is unclear if OTUs or ASVs are being presented, although based on the pipeline presented ASVs would be assumed. Please add information on the sequence percent similarity used for generating OTUs or ASVs.

Author Response: We apologize for the lack of clarity. We did only use ASVs for our downstream analyses not OTUs. For our richness analysis, we mistakenly used OTUs but mean richness or number of unique ASVs. We have made these changes throughout the document now and figures.

There is inconsistencies on how the p-values are presented. Would recommend using $p < 0.0001$ for those values presented in the text with scientific notation.

Author Response: We have now corrected these issues.

I would encourage the authors to proofread the manuscript again for grammar errors. For instance, line 67, 'was' should be 'were' and line 521, 'the' should be 'that'.

Author Response: We have done our best to identify and correct these grammar issues.

Lines 426 to 435. Refrain from discussing the study findings related to the literature in the results.

Author Response: We have now corrected these issues.

Lines 520 to 521. This information should be reported in the results and could even be even added to Figure 2.

Author Response: Thank you. We have now corrected these issues.

Lines 717. Term dose-dependent is used, however, this study design is not able to evaluate does dependent effects.

Author Response: Thank you. We have now removed this word.

While different intervention and control gummies were used that differed in their carbohydrate sources (tapioca vs. rice syrup), it is very unlikely that these differences would alter the microbiota unless one of the syrups contained IMO or dietary fibers like resistant maltodextrin. However, the gummies do/may differ in other ingredients such as the carnauba wax and pectin. Therefore, the limitation would be that the control is not a match formulation without the vitamin D.

Author Response: Thank you. We have now revised this statement “There were also differences in sugars (tapioca vs rice syrup) between the two formulations of intervention and placebo gummies, though this likely did not impact the microbiome significantly since this difference was not attributed to microbiota-accessible carbohydrates.”

Figure 5. Unclear in legend which timepoint is being used for the analysis.

Author Response: We have now completely revised this figure and figure legend.

Re: Spectrum00083-24R1 (Randomized control trial of moderate dose vitamin D alters microbiota stability and metabolite networks in healthy adults)

Dear Dr. Leigh Greathouse:

Thank you for the privilege of reviewing your work. As you can see from comments from reviewer #2 below, there are just a few small sections of the manuscript that need to be updated before the manuscript can be accepted.

Revision Guidelines

Sincerely,
Jennifer Auchtung
Editor
Microbiology Spectrum

Reviewer #2 (Comments for the Author):

In the revised version of the manuscript, the authors have addressed sufficiently my previous comments and I appreciate their effort to discuss in detail some of my concerns. Even though extra analysis has been performed by the authors to address the comments, the manuscript has not always been updated to include these changes and new analysis. Below I note some of these points together with some other minor comments, but I would advise the authors to have an additional look for points that I might have missed.

Line 329: 'Spearman rank correlation was performed to identify taxonomic changes (enrichment/depletion) over time.' I believe that sentence does not hold any more as you later performed a different analysis.

Line 377: Figure 3C is not depicting a PCoA anymore. You need to rephrase here according to the new figure/analysis.

Line 394: Here again you need to update it to correspond with the current analysis/figures.

Lines 409-410 and lines 427-430: The terms 'acute' and 'persistent' are used differently in the two sentences. The first sentence is not very explanatory as 'at least one time point' and 'multiple timepoints' are implying the same.

Lines 466-473: This part fits better in the discussion.

Line 628: RNA-seq is method used for metatranscriptomics not for metagenomics. You should adjust the sentence.

Figure 3D : the figure legend reads 'Principal Coordinate Analysis (PCoA) based on the centroid of Aitchison's Distance between days 7, 14, and 78 of each participant in both placebo and treatment group and comparison between the distance of each subjects' centroid from the mean (Wilcoxon's test).' Probably this was not changed when the figure changed. You need to adjust to correspond to new figure.

Figure 5: A line between timepoint 7 and 14 is missing making the figure slightly hard to read. Additionally, you could include the points/regression for the timepoints that it is not significant to make the figure more complete.

Reviewer #3 (Comments for the Author):

The manuscript by Wyatt et al., describes the effect of 4000 UI vitamin D supplementation on the fecal microbiota. The authors have adequately addressed the concerns noted by the reviewers; I have no further concerns.

Response to Reviewers

Reviewer #2:

In the revised version of the manuscript, the authors have addressed sufficiently my previous comments and I appreciate their effort to discuss in detail some of my concerns. Even though extra analysis has been performed by the authors to address the comments, the manuscript has not always been updated to include these changes and new analysis. Below I note some of these points together with some other minor comments, but I would advise the authors to have an additional look for points that I might have missed.

Line 329: 'Spearman rank correlation was performed to identify taxonomic changes (enrichment/depletion) over time.' I believe that sentence does not hold any more as you later performed a different analysis.

Author response: *We appreciate the reviewer pointing out this error in our reporting. We have now amended this sentence to instead say that a pairwise ANOVA, DESeq2, and ALDEx2 were performed to identify taxonomic changes (enrichment/depletion) over time.*

Line 377: Figure 3C is not depicting a PCoA anymore. You need to rephrase here according to the new figure/analysis.

Author response: *We have updated Figure 3C to reflect the appropriate analysis, however, the text in line 377 we believe still reads correctly.*

Line 394: Here again you need to update it to correspond with the current analysis/figures.

Author response: *We appreciate the reviewer pointing out this error in our reporting. We have now amended this sentence to instead say that a pairwise ANOVA, DESeq2, and ALDEx2 were performed to identify taxonomic changes (enrichment/depletion) over time.*

Line 398:

Author response: *We have now revised this sentence to indicated that between Day 0 and Days 7, 14 and 78 after vitamin D3 intervention (pairwise ANOVA, DESeq2 and ALDEx2).*

Lines 409-410 and lines 427-430: The terms 'acute' and 'persistent' are used differently in the two sentences. The first sentence is not very explanatory as 'at least one time point' and 'multiple timepoints' are implying the same.

Author response: *We have now revised the use of these terms to provide a more consistent usage and a more detailed explanation of their terminology. Line 419*

Lines 466-473: This part fits better in the discussion.

Author response: *Thank you for the suggestions. We have now placed it the discussion section, line 577*

Line 628: RNA-seq is method used for metatranscriptomics not for metagenomics. You should adjust the sentence.

Author response: We have now revised the sentence. “Furthermore, we were limited in this study to only 16S rRNA sequencing as opposed to a more comprehensive metagenomic/metatranscriptomic (e.g., WGS or RNA-seq), and metabolomic analyses.”

Figure 3D : the figure legend reads 'Principal Coordinate Analysis (PCoA) based on the centroid of Aitchison's Distance between days 7, 14, and 78 of each participant in both placebo and treatment group and comparison between the distance of each subjects' centroid from the mean (Wilcoxon's test)!' Probably this was not changed when the figure changed. You need to adjust to correspond to new figure.

Figure 5: A line between timepoint 7 and 14 is missing making the figure slightly hard to read. Additionally, you could include the points/regression for the timepoints that it is not significant to make the figure more complete.

Author response: Thank you, we have now revised the figure captions.

Revised Captions

Figure. 3 Vitamin D₃ intervention alters microbiome β -diversity and homogenizes microbial composition. (A) The comparison of β -diversity using Aitchison's distance between the subjects within placebo group and within treatment group at four timepoints- Day 1, 7, 14, and 78 (Wilcoxon's paired test). (B) Principal Coordinate Analysis (PCoA) comparison of community composition between groups using PERMANOVA (Adonis2, $n=1000$) between all samples from days 7-14 and 14-78 of each participant. (C) Dispersion analysis based on the Aitchison's Distance between days 7, 14, and 78 of each participant in both placebo and treatment group (PERMANOVA, $n=1000$). (D). Microbiome stability (inverse Aitchison's Distance) between consecutive timepoints for each participant in the placebo and treatment group with percent changes in serum 25(OH)D levels (Generalized Linear regression). Each vertical line of dots represents one participant and each dot within this line is the mean of the Aitchison's distance between days 1-7, 7-14, and 14-78.

Figure 4. Vitamin D₃ induces changes in specific taxa that are both acute and persistent. Heatmaps showing changes in microbial taxonomic abundance at the genus level across the days for the placebo A) or treatment B) group (aggregate of features or ASVs from pairwise ANOVA, DESeq2, and ALDEx2). The three rows represent taxonomic changes from days 1-7, 7-14, and 14-78 in order. Red represents a decrease in microbial population while green represents an increase in microbial population (*; $p<.05$, **; $p<.01$). C) The changes in relative abundances of the top temporally changing microbes of the vitamin D group. Principal response curves (PRC) were used to determine the contribution of each microbe towards the collective microbiome change over time in vitamin D microbiomes, where day 1 was used as the PRC baseline to compare the collective microbiota changes in days 7 and 14. PRC species scores were used to determine the top positively and negatively influenced temporal microbial changes in the vitamin D group. Y axis represents the mean relative abundance calculated for each organism while the x-axis depicts sampling days. In addition, the relative abundance of day 78 is added separately into each plot to showcase the persistence of these temporal changes beyond the original experimental timeframe. Placebo trendline shows the general trend of each plotted organism for the placebo group across the days 1 to 78.

Figure 5. Vitamin D₃ levels are correlated with change in the abundances of specific taxa. The graph shows the taxa with a significant correlation (Pearson's, $p<0.05$) between the change of their abundance (Centred Log-Ratio) for Days 7, 14, and 78 from Day 1 and the change in serum Vitamin D level in the subjects of the treatment group.

New Supplementary Figure. Linear Regression and correlation models (Spearman's) of the body composition variables, physical activity variables, sleeping behavior variables, dietary score, age, and

Vitamin D (baseline, post-intervention, and % change) levels with Microbiome Stability (as described in Figure 3 (D)) (A), Vitamin D change % (B) and Vitamin D level-post intervention.

Re: Spectrum00083-24R2 (Randomized control trial of moderate dose vitamin D alters microbiota stability and metabolite networks in healthy adults)

Dear Dr. Leigh Greathouse:

Your manuscript has been accepted, and I am forwarding it to the ASM production staff for publication. Your paper will first be checked to make sure all elements meet the technical requirements. ASM staff will contact you if anything needs to be revised before copyediting and production can begin. Otherwise, you will be notified when your proofs are ready to be viewed.

Sincerely,
Jennifer Auchtung
Editor
Microbiology Spectrum